# Development and evaluation of processes affecting simulation of diel fine particulate matter variation in the GEOS-Chem model

Yanshun Li[1], Randall V. Martin[1,2], Chi Li[1], Brian L. Boys[2], Aaron van Donkelaar[1], Jun Meng[3], Jeffrey R. Pierce[4]

[1]Department of Energy, Environmental & Chemical Engineering, Washington University in St. Louis, St. Louis, Missouri, USA
[2]Department of Physics and Atmospheric Science, Dalhousie University, Halifax, Nova Scotia, Canada
[3]Air Quality Research Division, Environment and Climate Change Canada, Toronto, Ontario, Canada
[4]Department of Atmospheric Sciences, Colorado State University, Fort Collins, Colorado, USA

*Correspondence to*: Yanshun Li (yanshun.li@wustl.edu)

**Abstract.** The capability of chemical transport models to represent fine particulate matter ($PM_{2.5}$) over the course of a day is of vital importance for air quality simulation and assessment. In this work, we used the nested GEOS-Chem model at $0.25° \times 0.3125°$ resolution to simulate the diel (24 h) variation in $PM_{2.5}$ mass concentrations over the United States (US) in 2016. We evaluate the simulations with in situ measurements from a national monitoring network. Our base case simulation broadly reproduces the observed morning peak, afternoon dip and evening peak of $PM_{2.5}$, matching the timings of these features within 1-3 hours. However, the simulated $PM_{2.5}$ diel amplitude in our base case was 106% biased high relative to observations. We find that temporal resolution of emissions, subgrid vertical gradient between surface model level center and observations, as well as biases in boundary layer mixing and aerosol nitrate are the major causes for this inconsistency. We applied an hourly anthropogenic emission inventory, converted the $PM_{2.5}$ mass concentrations from model level center to the height of surface measurements by correcting for aerodynamic resistance, adjusted the boundary layer heights in the driving meteorological fields using aircraft observations, and constrained nitrate concentrations using in situ measurements. The bias in the $PM_{2.5}$ diel amplitude was reduced to -12% in the improved simulation. Gridded hourly emissions rather than diel scaling factors applied to monthly emissions reduced biases in simulated $PM_{2.5}$ overnight. Resolving the subgrid vertical gradient in the surface model level aided capturing timings of $PM_{2.5}$ morning peak and afternoon minimum. Based on the improved model, we find that the mean observed diel variation in $PM_{2.5}$ for the US is driven by 1) building up of $PM_{2.5}$ by 10% in early morning (4:00 - 8:00 local time, LT) due to increasing anthropogenic emissions into a shallow mixed layer, 2) decreasing $PM_{2.5}$ by 22% from mid-morning (8:00 LT) through afternoon (15:00 LT) associated with mixed layer growth, 3) increasing $PM_{2.5}$ by 30% from mid-afternoon (15:00 LT) though evening (22:00 LT) as emissions persist into a collapsing mixed layer, and 4) decreasing $PM_{2.5}$ by 10% overnight (22:00 - 4:00 LT) as emissions diminish.

## 1 Introduction

Airborne fine particulate matter ($PM_{2.5}$) affects human health (GBD 2019 Risk Factor Collaborators, 2020), visibility (Malm et al., 1994; Li et al., 2016) and the climate system (Pörtner et al., 2022). Accurately representing the diel $PM_{2.5}$ variation, its variation over the course of a day, is essential for exposure assessment, air quality modeling and relating

PM$_{2.5}$ concentrations at a specific time of day to daily averages (van Donkelaar et al., 2010; Manning et al., 2018). Ground-level observations have revealed similar bimodal diel PM$_{2.5}$ variations across the world, in which the mass concentrations typically peak in morning and late evening, with minima near daybreak and late afternoon (Manning et al., 2018). How well chemical transport models (CTMs) reproduce this variation has not been fully investigated.

Previous modelling studies over major anthropogenic source regions found mixed levels of skill in resolving diel
PM$_{2.5}$ variation. CTMs generally well capture the observed mid-morning and late evening peaks in PM$_{2.5}$ (Tessum et al., 2015; Bessagnet et al., 2016; Du et al., 2020). The peak in mid-morning is commonly attributed to enhanced anthropogenic emission activities and the peak in late evening ascribed to collapse of the planetary boundary layer (Zhao et al., 2009; Rattigan et al., 2010; Tiwari et al., 2013). Biases in simulated diel PM$_{2.5}$ variation have also been identified and investigated. Du et al. (2020) used the WRF-Chem model (Grell et al., 2005) with the MOSAIC (Model
for Simulating Aerosol Interactions and Chemistry) scheme and the CBM-Z (carbon bond mechanism) photochemical mechanism to simulate diel PM$_{2.5}$ variation over East Asia and found nighttime overestimation, possibly due to insufficient boundary layer mixing. Simulations from multiple CTMs in the EURODELTA III intercomparison study (Bessagnet et al., 2016) found notable underestimation of PM$_{2.5}$ concentrations in the afternoon over Europe. Lack of unspeciated organics and incomplete chemical mechanisms for the formation of secondary organic aerosols were
proposed as the driving forces.

Global anthropogenic emission inventories are generally available at monthly mean resolution (Janssens-Maenhout et al., 2015; Huang et al., 2017; McDuffie et al., 2020). These monthly inventories are often applied as is for a wide range of studies. Some national emission inventories (e.g., NEI) contain local species- and sector-specific diel variation. Such national information for a specific country has in some instances been applied to provide diel
information for global inventories in some models. There is need to explore the effects of these different approaches upon the diel variation in PM$_{2.5}$ concentrations.

The vertical extent of the lowest model level in CTMs is typically tens of meters above ground, while ground-based measurements are taken at around two meters. As subgrid vertical gradients exist between model level center and surface observations, CTM simulation and in situ measurements represent PM$_{2.5}$ at different altitudes. This so-called
vertical representativeness difference can affect model evaluation. Previous modeling studies have estimated subgrid vertical gradients in HNO$_3$ and O$_3$ within the first model level using dry deposition velocity and aerodynamic resistance (Zhang et al., 2012; Travis and Jacob, 2019). How such differences in vertical representation affect simulated diel PM$_{2.5}$ has not been investigated.

Aerosol dry deposition, defined as the removal of aerosols by gravitational settling, by Brownian diffusion, or by
impaction and interception resulting from turbulent transfer (Beckett et al., 1998), is an important sink process. Recent investigations have examined developments to the dry deposition scheme used in CTMs. Petroff and Zhang (2010) developed a sized-resolved particle dry deposition scheme with a new surface resistance parameterization by simplification of a one-dimensional aerosol transport model. Kouznetsov and Sofiev (2012) proposed a comprehensive particle dry deposition scheme which accounts for physical properties of the air flow, surface and depositing particles.
Zhang and Shao (2014) improved the modeling of particle dry deposition on rough surfaces by treating gravitational

settling analytically and considering the roughness in particle diffusion and surface collection. Emerson et al. (2020) revised size-resolved particle dry deposition through constraining the surface resistances using particle flux observations. The impacts of recent updates on $PM_{2.5}$ mass concentrations and its diel variation remains unclear.

Aerosol nitrate, mainly formed chemically from ammonia and nitric acid, is an important component of $PM_{2.5}$. Previous studies reported aerosol nitrate as overestimated in models, including GEOS-Chem (Heald et al., 2012), PMCAMx (Fountoukis et al., 2011) and WRF-Chem (Tuccella et al., 2012). Uncertainties in the heterogeneous uptake coefficient of $N_2O_5$ and $NO_2$, dry deposition velocity of nitric acid, and nighttime boundary layer has been investigated as potential factors causing the overestimation (Miao et al., 2020; Zhai et al., 2021; Travis et al., 2022). The overprediction of nitrate in GEOS-Chem was found most prominent during the night (Travis et al., 2022), which can affect the diel variation of $PM_{2.5}$.

In this work, we use the GEOS-Chem CTM, initially described by Bey et al. (2001), to investigate the diel variation in simulated $PM_{2.5}$. We focus on the US in 2016. In Sect. 2, we introduce the GEOS-Chem model and the configurations of our base simulation. In Sect. 3, we describe the in situ measurements of $PM_{2.5}$. The rest of the paper is organized by themes, each of which contains its own methodology, results and discussions. In Sect. 4, we evaluate and identify biases of the simulated diel $PM_{2.5}$ variation in our base GEOS-Chem simulation. Multiple physical and chemical processes affecting the diel $PM_{2.5}$ simulation are explored in Sect. 5 by developing the model and conducting sensitivity simulations, based on which we describe the revised diel simulation with discussions in Sect. 6. Sect. 7 concludes this study.

**2 The GEOS-Chem model and the base simulation**

**2.1 General description**

We use the GEOS-Chem chemical transport model version 12.6.0 (www.geos-chem.org) driven by the GEOS-5 Forward Processing (GEOS-FP) assimilated meteorology from the NASA Global Modeling and Assimilation Office (GMAO) to examine the factors controlling the diel $PM_{2.5}$ mass variations. Prior applications of the model to $PM_{2.5}$ studies include but are not limited to evaluating and improving mechanisms affecting $PM_{2.5}$ (Zheng et al., 2015; Marais et al., 2016; Song et al., 2021; Travis et al., 2022), source attribution (Meng et al., 2019; McDuffie et al., 2021; Pai et al., 2022), assessments of the effects of horizontal transport on local air quality (Lang et al., 2012; Zhang et al., 2019; Xu et al., 2023) and exposure assessments (Kodros et al., 2016; van Donkelaar et al., 2021).

GEOS-Chem simulates detailed tropospheric aerosol-oxidant chemistry which includes the sulfate-nitrate-ammonium system (Park et al., 2004; Fountoukis and Nenes, 2007), black carbon (Wang et al., 2014), organic carbon, secondary organic aerosol (Pai et al., 2020), mineral dust (Fairlie et al., 2007) and seasalt (Jaeglé et al., 2011). The so-called "simple" scheme (Kim et al., 2015) is used for simulating secondary organic aerosol (SOA). Absorption of radiation by brown carbon is implemented following Hammer et al. (2016). We use nested simulations over the US in 2016 at $0.25° \times 0.3125°$ over 47 vertical layers extending from the surface up to 0.1 hPa. The surface level extends from ground to about 120 meters. GEOS-FP is used for meteorological inputs, which includes hourly surface variables and

3-D variables at every 3 hours. A global simulation at $2° \times 2.5°$ is used to provide boundary conditions for the nested domain. The non-local scheme implemented by Lin and McElroy (2010) is used for boundary layer mixing.

In this work, we first evaluate the base simulation of GEOS-Chem (denoted as GC_Base in Table 1). We identify the biases of diel $PM_{2.5}$ variation in the base simulation by comparison with in situ observations. Then we develop different model components affecting $PM_{2.5}$ concentrations and conduct sensitivity simulations to explore the driving forces of

diel $PM_{2.5}$ variation. Sect. 2.2 and 2.3 introduce the emission configuration and default parameterization of dry deposition in GC_Base.

**Table 1. Summary of modifications made to base GEOS-Chem simulation to investigate diel $PM_{2.5}$ variation.**

| GEOS-Chem simulation | Temporal resolution of emissions | Vertical representativeness | Aerosol dry deposition | Boundary layer mixing | Nitrate constrained |
|---|---|---|---|---|---|
| GC_Base | NEI monthly | Lowest model level center | Default | Default | No |
| GC_Emis | NEI hourly | Lowest model level center | Default | Default | No |
| GC_Drydep | NEI hourly | Lowest model level center | Revised | Default | No |
| GC_2m | NEI hourly | Corrected to 2m | Revised | Default | No |
| GC_2m_PBLH | NEI hourly | Corrected to 2m | Revised | PBLH adjusted | No |
| GC_2m_PBLH_NIT | NEI hourly | Corrected to 2m | Revised | PBLH adjusted | Yes |

**2.2 Emissions configurations in GC_Base**

To investigate the impacts of anthropogenic emissions, we begin with the monthly version of the National Emission Inventory (NEI) in GC_Base instead of the default hourly version in the standard nested GEOS-Chem model over North America, which is consistent with most regions outside of the US where anthropogenic emissions at hourly resolution are often not readily available. We scale the NEI emissions from the base year of 2011 to 2016 using air pollutant emissions trend data provided by the US Environmental Protection Agency (EPA) (https://www.epa.gov/air-

emissions-inventories/air-pollutant-emissions-trends-data). Point sources in the NEI inventory are all vertically resolved, which mainly include large industrial facilities, power plants and airports. Nonpoint sources mainly include residential heating, transportation, commercial combustion and solvent use. We do not use the NEI 2016 inventory since that inventory is only available at monthly resolution in GEOS-Chem. For wildfires, we use GFED4 (Giglio et al., 2013) 3-hourly emissions. For dust, we use the hourly offline inventory developed by Meng et al. (2021).

## 2.3 Dry deposition parameterization in GC_Base

Dry deposition of PM$_{2.5}$ in our base GEOS-Chem simulation generally follows the Zhang et al. (2001) scheme (hence forth Z01), which parameterizes particle dry deposition velocities ($V_d$) by accounting for gravitational settling ($V_g$), aerodynamic resistance ($R_a$) and surface resistance ($R_s$), as shown in Eq. (1):

$$V_d = V_g + \frac{1}{R_a + R_s},$$  (1)

Gravitational settling represents the particle settling due to gravity. Aerodynamic resistance describes the turbulent transport of scalars within the surface layer. Surface resistance, as formulated in Eq. (2), quantifies particle-surface contact in close proximity to surfaces by Brownian diffusion ($E_b$), impaction ($E_{Im}$) and interception ($E_{In}$).

$$R_s = \frac{1}{\varepsilon_0 u_* (E_b + E_{Im} + E_{In}) R_1},$$  (2)

where $u_*$ denotes friction velocity, $R_1$ denotes a bounce correction term and $\varepsilon_0$ denotes an empirical coefficient. Brownian diffusion contributes to dry deposition through diffusion when particles are close to surface collectors. Impaction describes the direct collision of particles to surfaces due to inertia when particles move along the streamlines around collector surfaces. Interception represents the deposition by which particles are captured by surface collectors when their distances to the collectors are less than the radius of a single particle.

The standard GEOS-Chem dry deposition module used in our base simulation calculates dry deposition velocity ($V_d'$) following Eq. (3), where gravitational settling is ignored.

$$V_d' = \frac{1}{R_a + R_s},$$  (3)

The dry deposition of PM$_{2.5}$ includes sulfate, nitrate, ammonium, organics, black carbon, fine mode seasalt and fine mode mineral dust components. Information about particle size is important, as all terms in Eq. (1-3) are size-dependent except aerodynamic resistance $R_a$. The dry deposition module in the base GEOS-Chem simulation has inconsistencies with other GEOS-Chem modules that we address in Sect. 5.2. In the standard GEOS-Chem dry deposition module, fine mode mineral dust is considered in two size bins with mass-weighted mean diameters of 1.46 and 2.80 μm. Other components are each considered in a single size bin with mass-weighted mean dry diameters for sulfate, nitrate, ammonium, organics, black carbon and fine mode seasalt of 0.5 μm. Monodisperse size distributions are used for all size bins. The effect of hygroscopic growth on deposition is only considered for fine mode seasalt following Lewis and Schwartz (2006). We use the standard GEOS-Chem dry deposition module for our base simulation.

## 3 In situ measurements of PM$_{2.5}$

The in situ measurements from the United States Environmental Protection Agency's Air Quality System (AQS) are used to evaluate the GEOS-Chem simulations. There were 451 sites operating in 2016 across the US which provided hourly PM$_{2.5}$ concentrations using a Federal Equivalency Method (FEM). As depicted in Fig. 1, 66.3% of these FEM

sites are equipped with the Met One BAM-1020 Mass Monitor using Beta Attenuation, 10.0% with the Thermo Scientific 5014i/FH62C14-DHS Monitor using Beta Attenuation, 7.4% with the Thermo Scientific TEOM 1405-DF Dichotomous Monitor using FDMS Gravimetric and 6.5% with the Thermo Scientific 5030 SHARP Monitor using Beta Attenuation. These four types of FEM monitors are used for hourly analysis in this work. The other five types of FEM instruments, contributing less than 10% of all hourly measurements, are excluded to avoid risk of aliasing instrument-dependent and regionally dependent characteristics. Further detail about instrumentation is provided in supplemental Sect. S1. A small fraction (0.05%) of the FEM measurements exceeding ten times their standard deviation are indictive of strong fire contamination, present significant modulation on the regional diel variation pattern and are excluded as outliers from the focus of this study. Also shown in Fig. 1 are the additional 737 sites using Federal Reference Method (FRM) to measure 24-hour average $PM_{2.5}$ concentrations which significantly improve observational coverage of the US for the evaluation of spatial distribution of GEOS-Chem simulated $PM_{2.5}$. To compare with GEOS-Chem, each site is matched with the GEOS-Chem grid nearest box center. The FRM and FEM measurements used in this work are at 35±5% relative humidity (EPA, 2007; Thermo Fisher Scientific, 2013; EPA, 2021; EPA, 2023). To match the measurement RH, the GEOS-Chem $PM_{2.5}$ and its composition were calculated considering the corresponding hygroscopic growth following standard practice in GEOS-Chem (GEOS-Chem Aerosols Working Group, 2021).

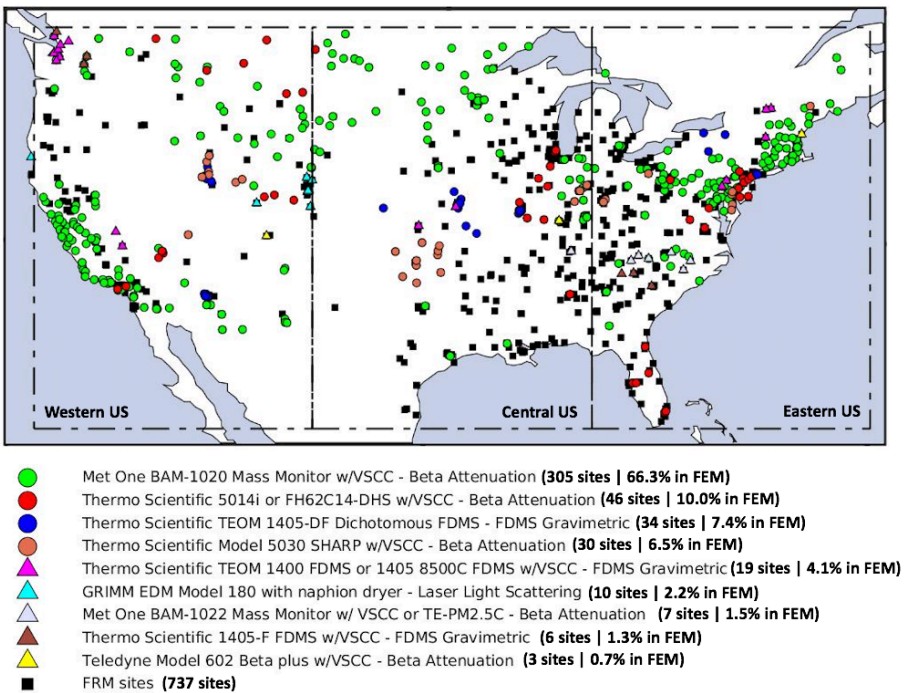

**Figure 1. Spatial distribution of the US EPA $PM_{2.5}$ measurements. Colored markers represent Federal Equivalency Method (FEM) sites equipped with different kinds of instruments which report hourly $PM_{2.5}$ concentrations. Black squares represent Federal Reference Method (FRM) sites which report 24-hour average $PM_{2.5}$.**

**4 Diel PM$_{2.5}$ variation in the base GEOS-Chem simulation and the FEM measurements**

We first examine the diel PM$_{2.5}$ variation in the base simulation. Fig. 2a shows annual-mean diel PM$_{2.5}$ variation across the US from the FEM in situ observations and the space and time co-located base GEOS-Chem simulation. The observed PM$_{2.5}$ exhibits a typical diel cycle consistent with previous work (Manning et al., 2018). Concentrations peak at 8am, diminish until late afternoon, increase in evening and remain elevated throughout the night. The base GEOS-Chem simulation broadly captures these features with their timings accurate within 1-3 hours. The simulated concentration decreases from morning to late afternoon then increases throughout the evening, consistent with the diel cycle of growth and collapse of the boundary layer. However, the simulated PM$_{2.5}$ is significantly overestimated at night, especially from midnight to early morning when the GEOS-Chem PM$_{2.5}$ increases beyond the standard deviation of the observations during which time the observations exhibit a slight decrease. The nighttime model overestimation leads to a 106% positive bias in the PM$_{2.5}$ diel amplitude, defined as the difference between the maximum and the minimum of the normalized diel concentration. The Root Mean Square Deviation (RMSD) of annual diel variation in PM$_{2.5}$ between the base simulation and the observations is 2.18 $\mu g/m^3$. The spatial distribution of PM$_{2.5}$ in the base GEOS-Chem simulation is discussed in supplemental Sect. S2.

We classify each FEM measurement and the corresponding GEOS-Chem simulation into urban and rural using the Global Rural-Urban Mapping Project (GRUMP) v1 (Balk et al., 2006) data at 30 seconds resolution. Results (Fig. S3) indicate that the observed diel variations of PM$_{2.5}$ in urban and rural areas across the US are highly consistent (r=0.97). Both urban and rural sites exhibit the same bi-modal patterns with PM$_{2.5}$ peaks near 8:00 LT and 21:00 LT, and minima near 4:00 LT and 16:00 LT. The PM$_{2.5}$ dips near 4:00 LT and 16:00 LT are deeper over urban regions than over rural regions, which may reflect stronger vertical mixing from the urban heat island effect (Travis et al., 2022). The consistency of diel PM$_{2.5}$ variation across urban and rural locations implies a dominant role from natural processes.

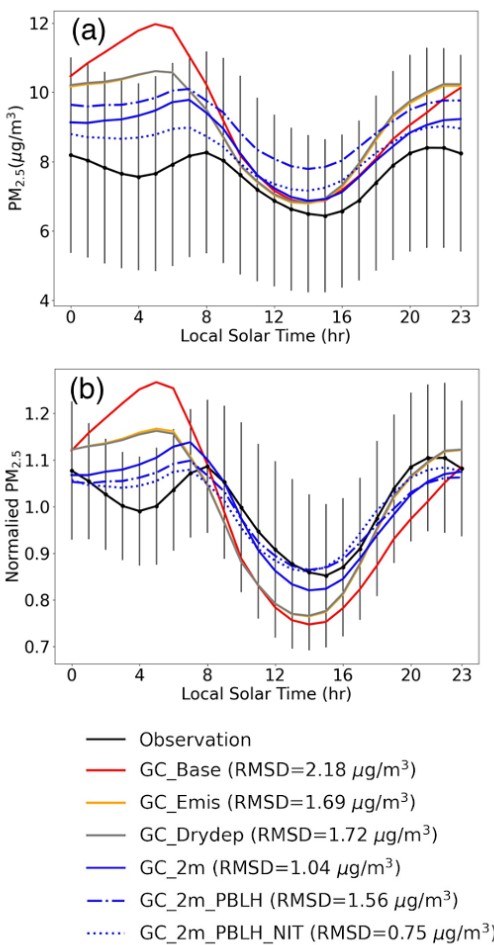

**Figure 2. (a) Annual mean diel PM$_{2.5}$ variation over the US in 2016. (b) Normalized annual mean diel PM$_{2.5}$ from GEOS-Chem (GC) sensitivity simulations over the US in 2016. Vertical lines indicate the spatial standard deviations of annual-mean PM$_{2.5}$ for the FEM measurements at each hour.**

Fig. 3a shows the annual-mean diel variation of PM$_{2.5}$ chemical composition in the base GEOS-Chem simulation for the contiguous US. Sulfate was the least variant component throughout the day. All other components exhibit notably higher concentration at night than during the day. The pronounced PM$_{2.5}$ accumulation overnight in the base case simulation is driven primarily by nitrate, of which the mass concentrations increase by 34.1% overnight (0:00 LT-6:00 LT). This is consistent with the reported overestimation of nighttime nitrate in GEOS-Chem by recent studies (Miao et al., 2020; Zhai et al., 2021; Travis et al., 2022). Concentrations of ammonium and SOA, which increased by 22.2% and 14.2% overnight (0:00 LT – 6:00 LT), contributed to the overnight PM$_{2.5}$ accumulation to a lesser extent. Except for dust, concentrations of all other components increase from midnight to early morning, indicating there are uniform drivers on PM$_{2.5}$ diel variation across composition.

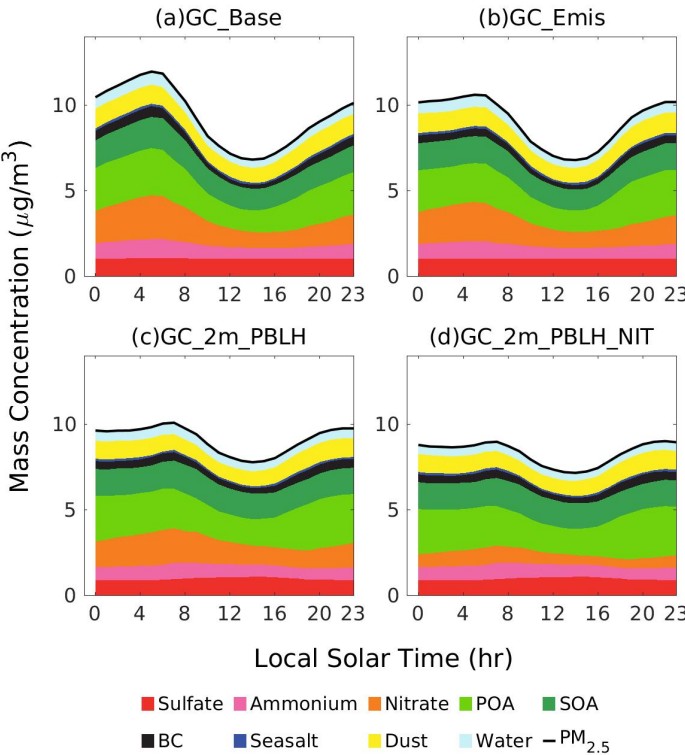

**Figure 3. Annual diel profiles of PM₂.₅ composition over the US in the GEOS-Chem simulations (Table 1). POA, SOA, BC refers to primary organic aerosol, secondary organic aerosol and black carbon respectively. All components represent dry mass. The aerosol water associated with sulfate, nitrate, ammonium, POA, SOA and seasalt is grouped into the water category.**

## 5 Development of processes affecting simulation of diel PM₂.₅

We develop and evaluate the processes affecting the simulation of diel PM₂.₅ variation in GEOS-Chem with particular attention to the driving forces of the nighttime bias. We focus on the temporal resolution of emissions, aerosol dry deposition, vertical representativeness, boundary layer mixing, dew formation, and nitrate as summarized in Table 1.

### 5.1 Impacts from the temporal resolution of emissions

We initially examine the temporal resolution of anthropogenic emissions as a source of the nighttime PM₂.₅ positive bias identified in Sect. 4. Fig. 4 shows the normalized mean diel emission profile for different species in the hourly version of the NEI inventory. Anthropogenic emissions are notably higher during the day than at night, with minima from midnight to early morning in the emission intensities for every primary species. The diel amplitude of $SO_2$ emissions is weakest, driven by persistent power plant emissions. $NH_3$ emissions have the strongest diel amplitude, driven by a temperature dependence for this predominantly agriculturally emitted species over the US (Zhang et al., 2018). Fig. S4 depicts the normalized mean emission strengths for species in Fig. 4 both seasonally and regionally. The early afternoon $NH_3$ peak is most prominent over the Central US in summertime, in accordance with the

temperature-dependant agricultural emissions of $NH_3$. Primary emissions of particulate organic carbon (OC) have a peak near l8:00 LT (Local Time), corresponding to more intense residential heating. The OC emissions in evening are strongest during winter, reflecting the seasonality of residential combustion activities (Li and Martin, 2018).

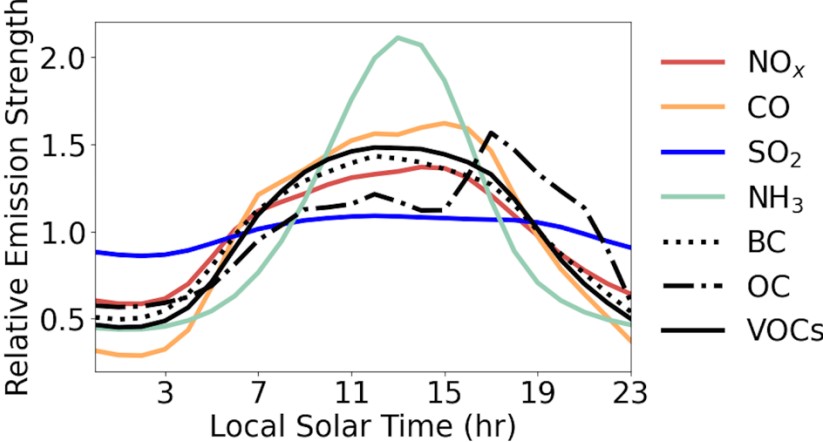

**Figure 4. Normalized mean diel emission profile for different species across the US.**

To evaluate the impacts from temporal resolution of emissions, we conduct a sensitivity simulation GC_Emis (Table 1) which replaces the monthly NEI in GC_Base with the hourly NEI. Fig. 2b shows that GC_Emis simulates a much weaker $PM_{2.5}$ accumulation from midnight to early morning relative to GC_Base, mainly due to the lower emission intensities of aerosol sources throughout the night in the NEI hourly inventory. In the evening, $PM_{2.5}$ in the GC_Emis simulation accumulates slightly faster than in the base case, reflecting the stronger emissions in daytime after applying the hourly inventory. The RMSD between GC_Emis diel $PM_{2.5}$ and the FEM observations decreases from 2.18 µg/m$^3$ in GC_Base to 1.69 µg/m$^3$, and the positive bias in the diel amplitude drops from 106% to 59%. In terms of composition (Fig. 3b), the average mass concentrations of BC and POA overnight (0:00 LT - 6:00 LT) decrease by 25.7% and 12.9%, contributing the most to the reduced overnight $PM_{2.5}$ accumulation. Sulfate concentrations overnight decrease by only 3.5% due to weak day-night contrast in $SO_2$ emissions. Nitrate and ammonium concentrations decrease by only 7.1% and 6.3%, reflecting the relatively minor role of primary emissions versus secondary production for these two species. In GC_Emis, nitrate still accumulates notably (by 23.1%) from 0:00 LT to 6:00 LT, acting as the major contributor of the $PM_{2.5}$ nighttime bias. Overall, the temporal resolution of emissions explains 44% of the bias in simulated diel amplitude. Daytime $PM_{2.5}$ is insensitive to changes in diel emission profiles. During the night, the impacts of emissions on $PM_{2.5}$ levels are more prominent, especially from midnight to early morning when the boundary layer is more stable. From this perspective, the slight overnight reduction of $PM_{2.5}$ in the FEM measurements is likely driven by the sharp decline in anthropogenic emissions.

The above analysis indicates the importance of using hourly emissions to simulate diel $PM_{2.5}$ variation. However, over most regions worldwide, only monthly emissions are available with crude diel scaling factors from specific regions as a possible proxy for hourly emissions. To assess the performance of such diel scalars in simulating diel $PM_{2.5}$, we conducted three supplementary sensitivity simulations in Table S2, in which sector or species-specific diel scaling

factors (Fig. S5) are applied to NEI and CEDS monthly emissions. Results (Fig. S6) show that the average PM$_{2.5}$ accumulation overnight (0:00 LT – 6:00 LT) among the supplementary cases is 2.6 times of that in GC_Emis, leading to stronger overestimation of PM$_{2.5}$ overnight. To optimize the model performance in simulating diel PM$_{2.5}$, hourly gridded emissions are preferred over using monthly emissions with scaling factors. Nevertheless, the diel emission profile does not fully explain the diel biases identified in Sect. 4. Other contributing factors exist.

**5.2 Impacts from the dry deposition parameterizations**

We explore dry deposition as the second potential source for the diel-varying biases in the base GEOS-Chem simulation. First, as described in Sect. 2.3, the dry deposition scheme in the base GEOS-Chem model does not account for gravitational settling $V_g$, which leads to systematic underestimation in particle dry deposition velocities. To improve on this missing consideration, we strictly follow Eq. (1) of Zhang et al. (2001), updating the gravitational settling term $V_g$ to be explicitly considered when deriving the deposition velocity (Eq. 1). Second, the parameterization of surface resistances (Eq. 2) in the base scheme was developed when few particle deposition measurements were available. Following recent observational evidence, Emerson et al. (2020) identified that the Brownian diffusion $E_b$ in Z01, as used in the standard GEOS-Chem model, is excessive while the contribution from interception $E_{In}$ is too weak. We update the surface resistances $R_s$ in GEOS-Chem by applying observationally-constrained Brownian diffusion $E_b$, impaction $E_{Im}$ and interception $E_{In}$ terms following observational evidence in Emerson et al. (2020). Formulations of $E_b$, $E_{Im}$, and $E_{In}$ are updated following Table 2.

**Table 2. Formulations for particulate gravitational setting ($V_g$), Brownian diffusion ($E_B$), interception ($E_{IN}$) and impaction ($E_{IM}$) used in the calculation of deposition velocity ($V_d$).**

| Resistance Model | $V_g$ | $E_B$ | $E_{IN}$ | $E_{IM}$ |
|---|---|---|---|---|
| Vd_Base | - | $Sc^{-\gamma}$ | $\frac{1}{2}(\frac{D_p}{A})^2$ | $(\frac{St}{\alpha + St})^2$ |
| Vd_Z01 | $V_g = \frac{\rho D_p^2 gC}{18\eta}$ | $Sc^{-\gamma}$ | $\frac{1}{2}(\frac{D_p}{A})^2$ | $(\frac{St}{\alpha + St})^2$ |
| Vd_Revised | $V_g = \frac{\rho D_p^2 gC}{18\eta}$ | $0.2Sc^{-2/3}$ | $\frac{5}{2}(\frac{D_p}{A})^{0.8}$ | $\frac{2}{5}(\frac{St}{\alpha + St})^{1.7}$ |

A: characteristic radius for interception in Zhang et al. (2001).

$C$: the Cunningham correction factor.

$D_p$: Particle diameter.

$g$: gravitational acceleration constant

$Sc$: the Schmidt number.

$St$: the Stokes number.

$V_g$: gravitational settling velocity.

α: LUC-specific constant used in the impaction efficiency in Zhang et al. (2001), where LUC represents land use classification.

$\gamma$: LUC-specific exponent used in the Brownian diffusion efficiency in Zhang et al. (2001), which ranges from 0.5 to 0.58.

$\rho$: density of particle

$\eta$: viscosity of air

Fig. 5a shows $V_g$ as a function of particle diameter for the base (Vd_Base) and revised (Vd_Revised) parameterizations,
as well as according to the Z01 scheme (Vd_Z01). Comparison of the Vd_Base and Vd_Z01 curves indicates that
inclusion of $V_g$ in the calculation of $V_d$ for the Vd_Z01 case substantially increases dry deposition velocities for
particles larger than 2 μm in diameter. The Vd_Revised curve indicates that implementing observational constraints
on the surface resistances shifts the minimum in $V_d$ to a particle diameter of around 0.1 μm, reflecting a weakened
Brownian diffusion term and an enhanced interception term. Emerson et al. (2020) found that the parameterized size
dependent particle dry deposition velocities are more consistent with observations after implementing these
observational constraints. To further evaluate the impact of particle $V_d$ on diel PM2.5, the representation of aerosol size
distributions in the dry deposition scheme of GEOS-Chem, including hygroscopic growth, must be considered.

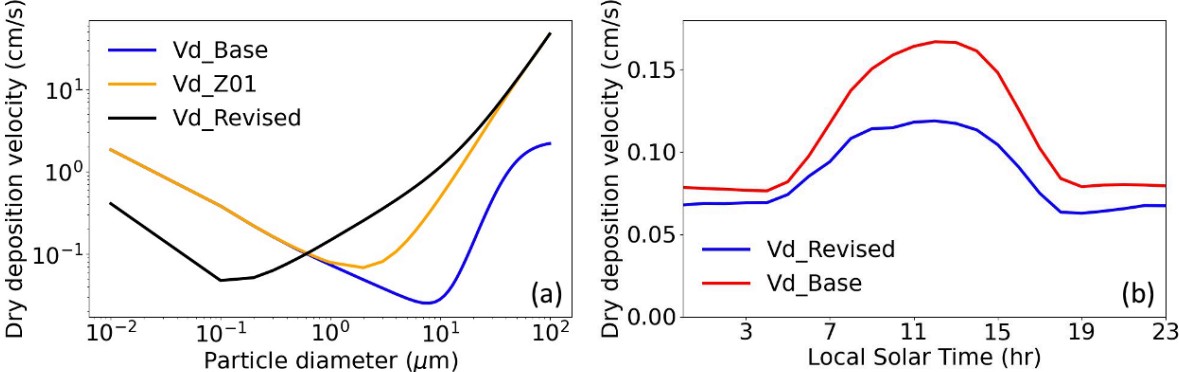

**Figure 5. (a) Size-resolved particle dry deposition velocities over grassland land type from GEOS-Chem. (b) Diel mean**
**dry deposition velocities for sulfate aerosol over the US in 2016. Vd_Base represents the default dry deposition scheme in**
**the base GEOS-Chem model (Eq. 3). Vd_Z01 includes the effect of gravitational settling on Vd_Base (Eq. 1). Vd_Revised**
**further implements the observational constrains on the surface resistance terms, as discussed in Sect. 5.2.**

As introduced in Sect. 2.3, the dry deposition scheme in the standard GEOS-Chem model assigns a single unreferenced
mass-weighted mean diameter to different PM2.5 components. We update the mass-weighted mean diameter for each
aerosol species dry deposited to be consistent with the sizes in the GEOS-Chem radiation module. We implicitly
consider aerosol size distributions based on mass conservation principles:

$$\int_0^\infty n(D_p) \cdot \frac{4}{3}\pi \left(\frac{D_p}{2}\right)^3 \cdot \rho \cdot V_d(D_p) dD_p = N \cdot V_d(D_p^*) \cdot \frac{4}{3}\pi \left(\frac{D_p^*}{2}\right)^3 \cdot \rho \,, \tag{4}$$

where $D_p$ denotes particle diameter, $n(D_p)$ represents the particle number size distribution, $\rho$ denotes the particle
density, $V_d(D_p)$ denotes the size-dependent particle dry deposition velocity, $N$ denotes the total particle number
concentration integrated across the aerosol size distribution, $D_p^*$ denotes the mass-weighted mean dry diameter for a
specific aerosol species and $V_d(D_p^*)$ denotes the dry deposition velocity of a particle with diameter of $D_p^*$. The size
distribution for each PM2.5 component is from Latimer and Martin (2019). The updated mass-weighted mean dry

diameter for sulfate, nitrate, ammonium and organic aerosols is 0.17 μm, for fine mode seasalt is 0.23 μm, and for the fine mode mineral dust in two size bins are 0.67 μm and 2.49 μm.

The standard GEOS-Chem dry deposition module only considers the hygroscopic growth of fine mode seasalt. Omitting hygroscopicity for other $PM_{2.5}$ components may lead to biases in the simulated dry deposition velocities and thus affect the diel variation of $PM_{2.5}$. Here we implement hygroscopic growth in the dry deposition parameterization for sulfate, nitrate, ammonium (SIA) and organic components (OA) of $PM_{2.5}$ by application of a κ-Kohler growth function to the mass-weighted mean dry diameters (Petters and Kreidenwei 2007, 2008, 2013; Latimer and Martin,

2019). Dust and black carbon are treated as hydrophobic. The κ-Kohler growth factor is calculated as:

$$GF = (1 + \kappa \frac{RH}{100-RH}) \,, \qquad\qquad\qquad\qquad (5)$$

The hygroscopicity parameter κ for SIA is set as 0.61 and for OA is set as 0.1 (Latimer and Martin, 2019). Efflorescence transitions are considered for the SIA components (Latimer and Martin, 2019). For fine mode seasalt, we continue to use the growth function from Lewis and Schwartz (2006).

Taking the sulfate component in $PM_{2.5}$ as an example, Fig. 5b presents the combined impacts of all the updates above on the diel dry deposition velocities. Implementation of gravitational settling and hygroscopic growth tends to increase the sulfate dry deposition velocity, compensating for the lower revised aerosol dry deposition velocities, mainly due to the revised scheme using a smaller mass-weighted mean dry diameter. The reductions of deposition velocity in the revised case are more prominent during daytime, when the size-dependent surface resistances dominate the dry

deposition processes. In the revised profile, from midnight to early morning (0:00 LT – 6:00 LT), the dry deposition velocities are 10.4% higher than those in the evening (18:00 LT-0:00 LT), reflecting the stronger aerosol hygroscopic growth due to higher relative humidity. We evaluate the impacts on simulated diel $PM_{2.5}$ masses in GEOS-Chem as GC_Drydep simulation (Table 1) which adds all the deposition updates to GC_Emis. Fig. 2b shows that the diel $PM_{2.5}$ masses simulated by GC_Drydep and GC_Emis are almost identical. The insensitivity of diel variation of $PM_{2.5}$ to dry

deposition updates implies that the diel $PM_{2.5}$ biases identified in Sect. 4 are unlikely to be caused by the uncertainty of the GEOS-Chem dry deposition module.

**5.3 Impacts from the vertical representativeness differences between model and observations**

The third possible contributor to the $PM_{2.5}$ nighttime biases that we consider is the vertical representativeness difference between the model and observations. Given the vertical extent of the lowest model level (120 m), simulated

concentrations represent an average over a greater vertical extent than the typical height of FEM measurements of about 2 meters. This difference in vertical representation may be especially problematic for model-measurement comparison during periods of diabatic stability resulting in strong near-surface concentration gradients. Vertical concentration gradients within 120 m of the surface have been widely observed for aerosol species in previous field campaigns (Sievering et al., 1994; Prabhakar et al., 2017; Franchin et al., 2018). Sievering et al. (1994) measured the

vertical profiles of aerosols over the Bayerischer Wald National Park in Germany using filter pack sampling, reporting 2 m concentrations lower than at 51 m for nitrate (51%), ammonium (81%) and sulfate (81%). In the Utah Winter Fine Particulate Study, the $PM_{2.5}$ concentrations measured by three ground sites at Logan, Cache, Salt Lake Valley

and the Utah Valley were around 70% of those at around 50 meters measured by aircraft (Franchin et al., 2018). Thus, the PM$_{2.5}$ simulated by GEOS-Chem is intrinsically different from the FEM in situ measurements because of the mismatch of vertical sampling location.

To evaluate the impact of these vertical representativeness differences, we developed the GC_2m simulation (Table 1), in which PM$_{2.5}$ from the lowest model level of the GC_Drydep simulation is adjusted to the height of the FEM measurements (2 meters above ground). The conversion process quantifies the vertical concentration gradient of secondary PM$_{2.5}$ components by using the resistance-in-series formulation for dry deposition following previous studies (Zhang et al., 2012; Travis and Jacob, 2019). The mathematical formula is described in Eq. 6,

$$C(z_{2M}) = [1 - R_a(z_{2M}, z_{GBC})V_d(z_{GBC})]C(z_{GBC}) , \qquad (6)$$

where $C(z_{2M})$ and $C(z_{GBC})$ represent the concentrations at measurement height of 2 meters and the grid-box-center of the GEOS-Chem surface layer (around 60 meters) respectively, $R_a(z_{2M}, z_{GBC})$ represents the aerodynamic resistances between the measurement height and the grid-box-center, $V_d(z_{GBC})$ represents the dry deposition velocity. $R_a(z_{2M}, z_{GBC})$ is calculated using the Monin-Obukhov similarity theory:

$$R_a(z_{2M}, z_{GBC}) = \int_{z_{GBC}}^{z_{2M}} \frac{\Phi(\zeta)}{ku^*\zeta} d\zeta , \qquad (7)$$

where $\zeta = z/L$. $L$ denotes the Monin-Obukhov length which is determined by surface momentum fluxes and sensible heat. $\Phi$ represents a function of stability described by Businger et al. (1971). $k$ represents the von Karman constant and $u^*$ represents the friction velocity. The method requires a boundary condition of zero concentration at ground. Thus, it is only applied to secondary PM$_{2.5}$ components, not primary components with surface emission fluxes. The correction method described by Eq. 6 and Eq. 7 does not account for the impacts of relative humidity (RH) and temperature (T) differences between the lowest model level and 2m on thermodynamic partitioning of sulfate-nitrate-ammonium (SNA) aerosol. Nevertheless, by conducting simulations of the Extended AIM Aerosol Thermodynamics Model (Wexler and Clegg, 2002) using GEOS-FP relative humidity (RH), Temperature (T) and GC_2m SNA composition at 2m and the lowest model level, we found the impacts are overall insignificant. Higher RH at 2m leads to SNA aerosol transition from solid to aqueous form and only slightly increases the ratio (<5%) of the partitioned aerosol phase in the SNA system, which usually occurs overnight.

Fig. 2b shows the normalized annual diel PM$_{2.5}$ variation of GC_2m across the US. Comparison of GC-Drydep and GC_2m indicates that the vertical correction effectively suppresses the excessive PM$_{2.5}$ levels from midnight to early morning and sustains the daytime concentration variation due to boundary layer mixing. The bias in diel amplitude of the corrected GC_2m PM$_{2.5}$ is reduced to 26% against the FEM observations. In terms of absolute concentrations, the average reduction from GC_Drydep to GC_2m is 1.01 µg/m$^3$ during 18:00 LT – 6:00 LT (nighttime), while that for 6:00 LT – 18:00 LT (daytime) is 0.11 µg/m$^3$. This day-night contrast is consistent with a previous DISCOVER-AQ field study (Prabhakar et al., 2017), in which the vertical gradient of nitrate aerosols measured by aircraft was significantly greater in a stable surface layer than in a turbulent surface layer. At night, under a stable boundary layer, surface resistances are suppressed due to weaker particle impaction and interception. Aerodynamic resistances then become relatively stronger with the resulting correction in Eq. 6 yielding a greater reduction of PM$_{2.5}$ concentrations.

During the day, as boundary layer mixing strengthens, surface resistances dominate over the aerodynamic resistances and the correction in Eq. 6 is weaker. Resolving the vertical representativeness differences enables the GEOS-Chem simulation to better capture the timings of the observed overall $PM_{2.5}$ morning peak and afternoon minimum across the US. In the GC_Drydep simulation, the $PM_{2.5}$ morning peak is three hours earlier than the FEM observations. After the vertical correction, in the GC_2m simulation, the morning peak appears only one hour ahead of the observations.

**5.4 Impacts from boundary layer height**

Planetary boundary layer height (PBLH) is investigated as the next possible source of the biases identified in Sect. 4. PBLH is closely related to boundary layer mixing, which significantly affects diel $PM_{2.5}$ (Du et al., 2020). We adjust the GEOS-FP planetary boundary layer height (PBLH) which used for driving GEOS-Chem by using the PBLH derived from the Aircraft Meteorological Data Reports (AMDAR) at 54 sites (Fig. S7) across the US (Zhang et al., 2020) as reference. The AMDAR PBLH is defined as the lowest level at which the bulk Richardson number exceeds a critical value of 0.5 (Zhang et al., 2020). The vertically resolved bulk Richardson number is calculated from vertical profiles of temperature, humidity and wind speed in the AMDAR dataset.

Fig. 6 shows the seasonal variation in PBLH. The observed PBLH from AMDAR shows similar diel variation across all seasons, which stays low from midnight to early morning, increases to a maximum in mid-afternoon, then decreases throughout rest of the day. In terms of absolute amplitude, the AMDAR PBLH are higher during spring and summer, mainly due to strong near-surface wind speed and intense solar radiation (Guo et al., 2016). The GEOS-FP reanalysis generally captures the diel variation of the AMDAR PBLH over all seasons, albeit with overestimates during daytime (7:00-19:00 LT), which is consistent as previous comparison studies (Millet et al., 2015; Zhu et al., 2016). The daytime overestimation in GEOS-FP PBLH is most likely due to excessive surface heating in the dataset. As reported in Millet et al., (2015), the daytime temperature at 2 meters in GEOS-FP was notably higher than that observed by ceilometer and the diel pattern of the bias in GEOS-FP temperature at 2 meters well matched that in PBLH. The average daytime AMDAR PBLH reaches a maximum in spring, slightly higher than that in summer, likely reflecting greater surface wind speed in spring than in summer according to the AMDAR observations, leading to greater turbulence and vertical mixing, and higher PBLH. GEOS-FP PBLH exhibits much higher values in summer than in spring. This inconsistency might be caused by stronger overestimation of GEOS-FP PBLH in summer introduced by excessive surface heating in the GEOS-FP dataset (Millet et al., 2015).






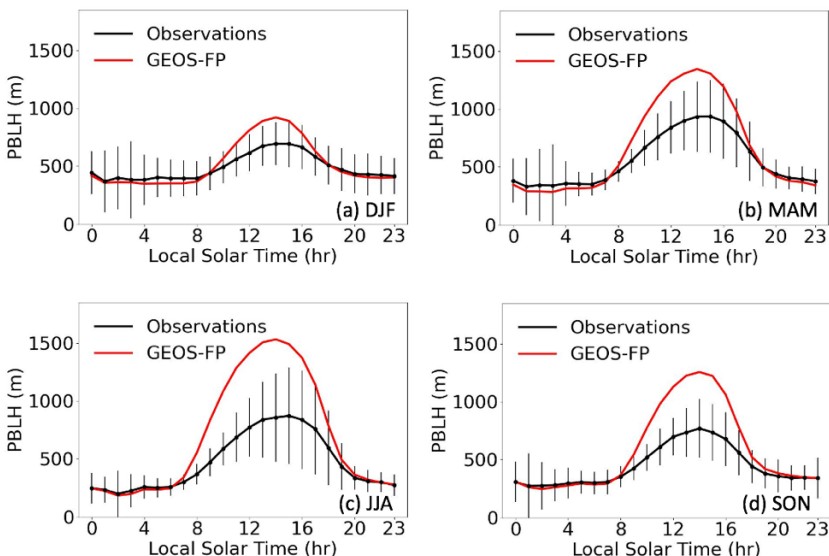

**Figure 6. Seasonal diel variation of AMDAR (observation-based) and GEOS-FP PBLH. Vertical bars indicate the spatial standard deviations of AMDAR PBLH.**

To quantify the impacts of the uncertainty in PBLH on modeled diel PM$_{2.5}$, we develop the GC_2m_PBLH simulation (Table 1) in which the GEOS-FP PBLH used in the GC_2m simulation is adjusted by the AMDAR observations. Specifically, we matched the hourly AMDAR and GEOS-FP PBLH over the US spatially and temporally, then derived US-averaged (0:00 LT – 23:00 LT) adjustment factors for different seasons following Eq. 8:

$$AF_{i,j} = \frac{\overline{PBLH_{AMDAR_{i,j}}}}{\overline{PBLH_{GEOS-FP_{i,j}}}}, \tag{8}$$

where $AF_{i,j}$ represents the PBLH adjustment factor for season i and hour j, $\overline{PBLH_{AMDAR_{i,j}}}$ represents the US-averaged AMDAR PBLH for season i and hour j, and $\overline{PBLH_{GEOS-FP_{i,j}}}$ represents the US-averaged GEOS-FP PBLH for season i, hour j. Implementing this adjustment scales the GEOS-FP PBLH to the same seasonal diel value as the AMDAR PBLH over the US. Applying these adjustment factors to the GEOS-FP PBLH, as shown in blue and dashed in Fig. 2b, reduces the absolute biases in simulated PM$_{2.5}$ diel amplitude against the FEM observations by 8%.

### 5.5 Impacts from dew formation

We also examined the possibility of dew formation as a potential process affecting the diel variation in PM$_{2.5}$. It was reported that the condensation process during the formation of dew involves removal of airborne particles from the atmosphere (Polkowska et al., 2008; Muskała et al., 2015). We considered whether the observed PM$_{2.5}$ decrease from midnight to early morning (Fig. 2) might be partly ascribed to this mechanism, and thus contribute to the overestimated nighttime PM$_{2.5}$. However, based on two lines of reasoning, we conclude here that dew formation is unlikely to significantly affect the diel PM$_{2.5}$ mass variations over the US. First, we examined co-located hourly RH and PM$_{2.5}$ mass concentrations at 37 sites in 2016 across the US. Fig. 7 shows four examples. We found no evidence of correlation of low PM$_{2.5}$ masses and high nighttime RH values (r=0.16/0.18/0.13/0.15 for Johnson, Kansas/Jackson,

Missouri/Summit, Ohio/Jefferson, Kentucky). Second, the decreases of PM$_{2.5}$ overnight are found sharpest in the Western US where the average relative humidity (RH) is lowest among all subregions, which indicates that dew formation at high RH condition is unlikely an important driving factor.

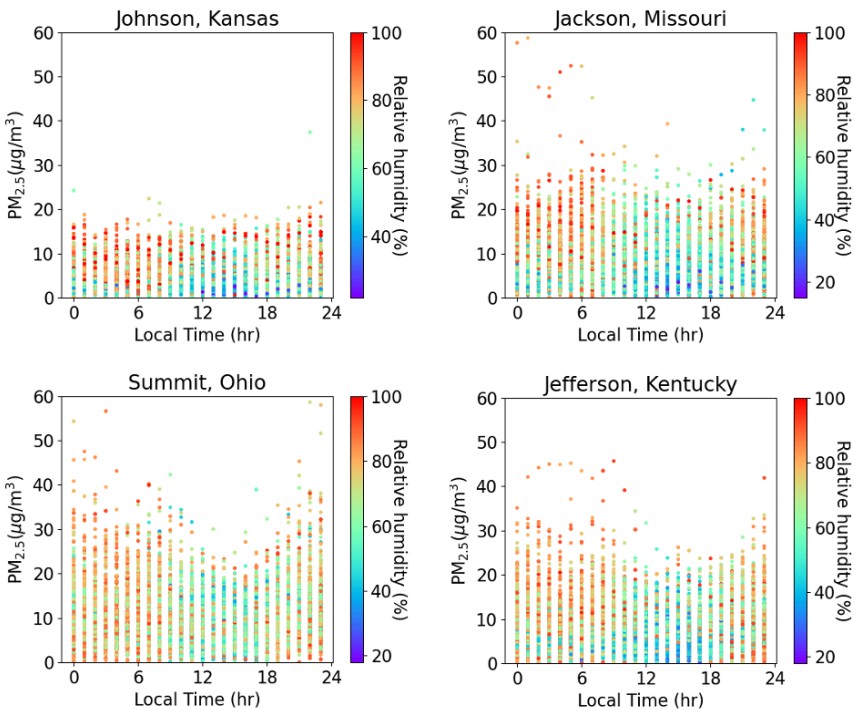


**Figure 7. Co-located relative humidity (RH) and PM$_{2.5}$ mass concentrations at four example sites. Each point represents the measured hourly PM$_{2.5}$ concentration at the measured hourly RH for each site. The RH measurements are provided by the NOAA Local Climatological Data (LCD) program. The PM$_{2.5}$ mass concentrations are provided by the EPA FEM sites.**

**5.6 Impacts from nitrate aerosols**

In Fig. 3b, hourly emissions reduce nighttime concentrations of nitrate and organics, primarily reflecting diminished nighttime emissions of NH$_3$, NO$_x$, and organic carbon. Accounting for vertical representativeness further reduces nighttime concentrations of nitrate (Fig. 3c), leading to reduced positive biases of 24h-averaged nitrate mass against in situ observations (Fig. S8). Nevertheless, positive nitrate biases remain in the GC_2m_PBLH simulation (Fig. S8),

which has been a long-standing issue in GEOS-Chem (Heald et al., 2012; Zhu et al., 2013). According to recent works (Miao et al., 2020; Zhai et al., 2021; Travis et al., 2022), uncertainties in aerosol uptake coefficient for N$_2$O$_5$ and NO$_2$, underestimated dry deposition of HNO$_3$ and overly shallow nighttime mixing layer are possible contributors. But none of these fully resolve the diel biases of nitrates in GEOS-Chem, indicating the biases are likely caused by misrepresentation in both chemistry and meteorology in the model. Following analyses by Travis et al. (2022) over

Seoul, Korea, we conducted sensitivity simulations (Sect. S3) and found N$_2$O$_5$ hydrolysis dominates the nighttime nitrate production (Fig. S9 and Fig. S10) in our simulations over the US, which is consistent with a previous work (Alexander et al., 2020). As shown in Fig. S9, turning off N$_2$O$_5$ hydrolysis largely reduces the PM$_{2.5}$ biases from

midnight to early morning and yields diel $PM_{2.5}$ variation highly consistent with observations. The bias in simulated
nitrate mass concentrations is also reduced by turning off $N_2O_5$ hydrolysis (Fig S8). The results indicate the $N_2O_5$
hydrolysis overnight might be excessive in the model. It is also possible that the performance of the simulation without
$N_2O_5$ hydrolysis on aerosols is an indicator of multiple chemical and physical processes affecting nitrate as explored
by Miao et al. (2020), Zhai et al. (2021) and Travis et al. (2022). While the full origins of the GEOS-Chem nitrate bias
remain unknown, we examine the effects on $PM_{2.5}$ of constraining nitrate concentrations by developing the
GC_2m_PBLH_NIT simulation, in which the modeled nitrate concentrations are halved from GC_2m_PBLH to better
represent the US average of in situ observations (Fig. S8). The bias of the diel amplitude of $PM_{2.5}$ in
GC_2m_PBLH_NIT against FEM observations is reduced to -12% (Fig. 2). The total aerosol water concentration
decreases by 12.7% in GC_2m_PBLH_NIT from GC_2m_PBLH as nitrate is reduced. These results motivate further
investigation of the nitrate bias in GEOS-Chem.

**6 Discussion of diel $PM_{2.5}$ variation in the final revised GEOS-Chem simulation (GC_2m_PBLH_NIT)**

Overall updating the temporal resolution of emissions, dry deposition parameterizations, boundary layer height,
resolving the vertical representative differences between model and observations and constraining nitrate notably
improves the diel variation of $PM_{2.5}$ in GC_2m_PBLH_NIT relative to GC_Base for both urban and rural regions (Fig.
S3) in a similar way. In the annual diel comparison averaged across the US (Fig. 2), the bias in the $PM_{2.5}$ diel amplitude
in GC_2m_PBLH_NIT (-12%) is significantly reduced relative to GC_Base (106%). The average observed $PM_{2.5}$
morning peak and afternoon minimum are at 8:00 LT and 15:00 LT respectively. GC_Base simulates them with biases
of -3 and -1 hours while GC_2m_PBLH_NIT agrees with observed timing within 1 hour. In addition to the average
comparison across the country, we further explore the performances over all FEM sites. Fig. 8 shows histograms of
the timing of the morning peak, of the afternoon minimum, and of the diel amplitude. At most FEM sites, GC_Base
tends to overestimate the $PM_{2.5}$ diel amplitude and simulates the $PM_{2.5}$ diel features too early. By correcting for the
vertical representativeness differences, using emissions with hourly temporal resolution, adjusting the GEOS-FP
boundary layer heights, and constraining nitrate concentrations, these biases are largely addressed in
GC_2m_PBLH_NIT with the distribution in the histogram better matching observations. The RMSD of diel $PM_{2.5}$
between GC_2m_PBLH_NIT and the FEM observations decreases from 2.18 to 0.75 $\mu g/m^3$. With reduced the 24-
hour averaged $PM_{2.5}$ concentration, GC_2m_PBLH_NIT also improves the agreements of annual-mean $PM_{2.5}$ against
the FEM/FRM measurements across the US (Sect. S2).

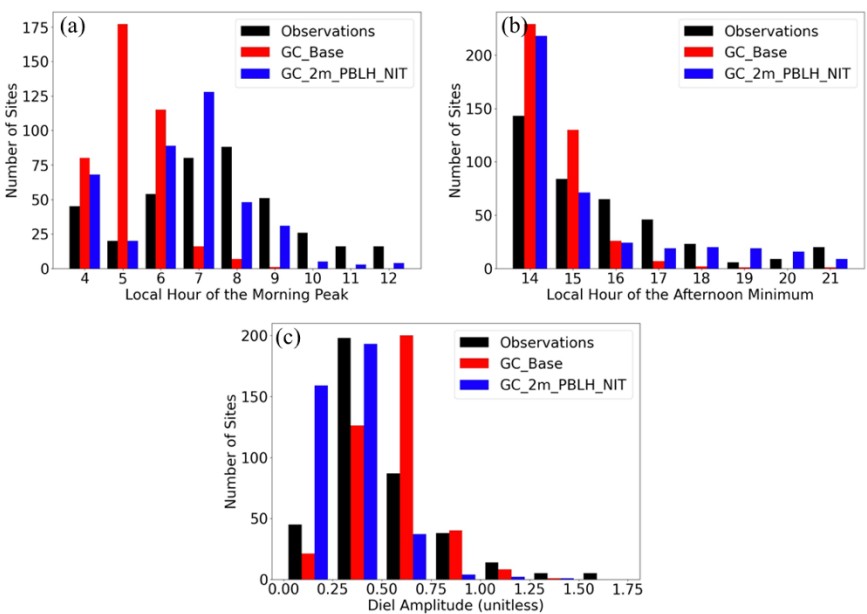

**Figure 8. Distribution of simulated and observed PM$_{2.5}$ features over the FEM sites. (a) Timing of morning peak. (b) Timing of afternoon minimum. (c) Diel Amplitude.**

Fig. 9 shows the diel variation of PM$_{2.5}$ in different seasons and subregions. The observed diel PM$_{2.5}$ variations are generally similar to the annual results across the country, suggesting consistent mechanisms controlling the local cycles. The observed PM$_{2.5}$ diel amplitude is smallest during summer, as the observed concentrations decrease more slowly from mid-morning to late afternoon than in other seasons. The GC_2m_PBLH_NIT simulation generally reproduces this summer minimum in diel amplitude, improving on GC_Base which simulates the minimum amplitude in winter, by reducing excess PM$_{2.5}$ at night, by reducing PM$_{2.5}$ precursor emissions and by accounting for vertical representativeness differences at night, by adjusting boundary layer height using aircraft observations and by constraining nitrate. Stronger photochemical production of PM$_{2.5}$ during daytime in summer than other seasons, also counteracts the ventilation by boundary layer mixing. The RMSD between GC_2m_PBLH_NIT and observed diel PM$_{2.5}$ improves on GC_Base for most seasons and subregions (Table S1).

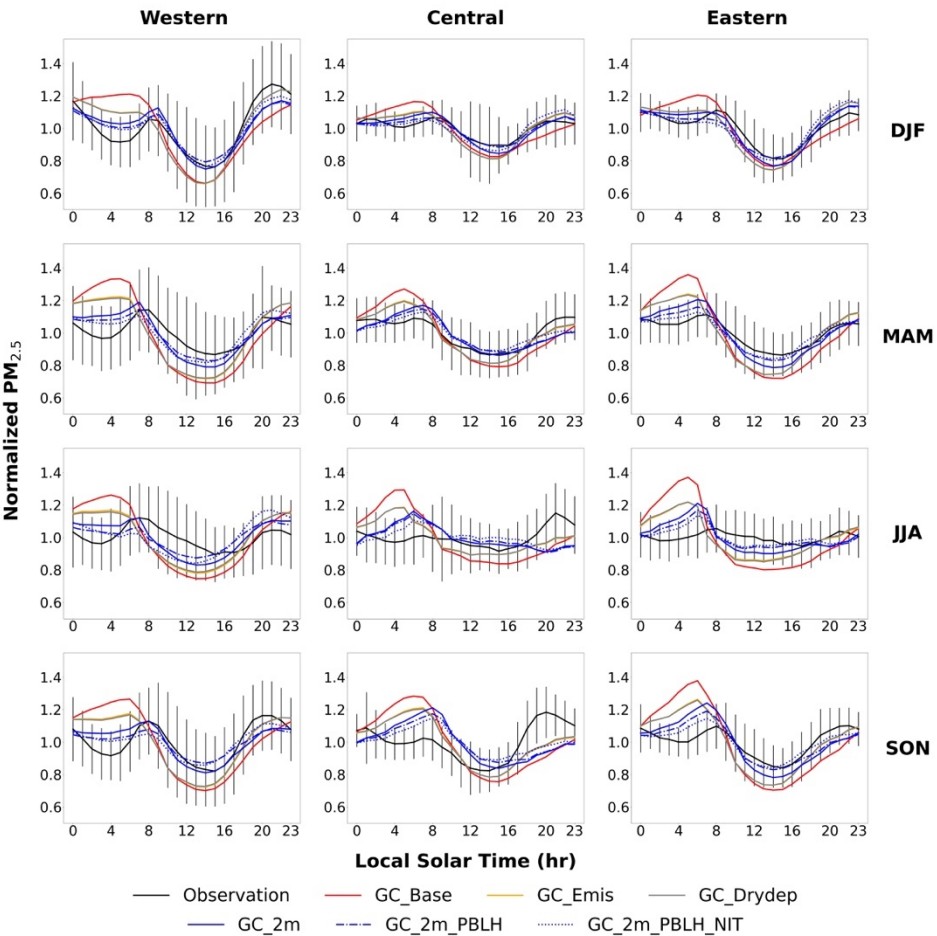

**Figure 9. Seasonal and regional diel profiles of GEOS-Chem PM$_{2.5}$ from different simulation designs (Table 1). Vertical lines indicate the spatial standard deviations of seasonal mean PM$_{2.5}$ for the FEM measurements at each hour in a certain subregion.**

Overall, we find that the driving forces of the typical diel PM$_{2.5}$ mass variation over the US reflects a complex interplay of PBL dynamics, emissions and photochemistry. The initial concentration peak in mid-morning occurs as combustion activities are emitted into a shallow mixed layer. Subsequent ventilation by vertical mixing dominates as the boundary layer develops, leading to a decrease of PM$_{2.5}$ until late afternoon despite enhanced photochemical production. The subsequent collapse at the mixed layer during sunset confines PM$_{2.5}$ emissions to the surface layer with a relative higher but diminishing concentration throughout the night as low nocturnal emissions foster a concentration minimum or flatness between midnight and early morning (Fig. 9). To further reveal the underlying driving forces, we focus on several example sites on which GC_2m_PBLH_NIT well reproduces the observed overnight PM$_{2.5}$ variation. Fig. 10 shows four example sites where the PM$_{2.5}$ concentrations overnight in the GC_Base simulation are substantially overestimated. By accounting for the hourly variation in anthropogenic emissions, in GC_Emis, the simulation starts to successfully reproduce the PM$_{2.5}$ decrease or flatness overnight. By further correcting for the vertical representativeness differences, adjusting boundary layer height and constraining nitrate, in GC_2m, GC_2m_PBLH

and GC_2m_PBLH_NIT, the simulations more closely represents the FEM measurements. These sensitivity
simulations reinforce that the internal driving forces of the PM$_{2.5}$ minimum or flatness from midnight to early morning
reflects a combination of the decrease of anthropogenic emissions by weaker anthropogenic activities, while resolving
the vertical representativeness differences between model and observations.

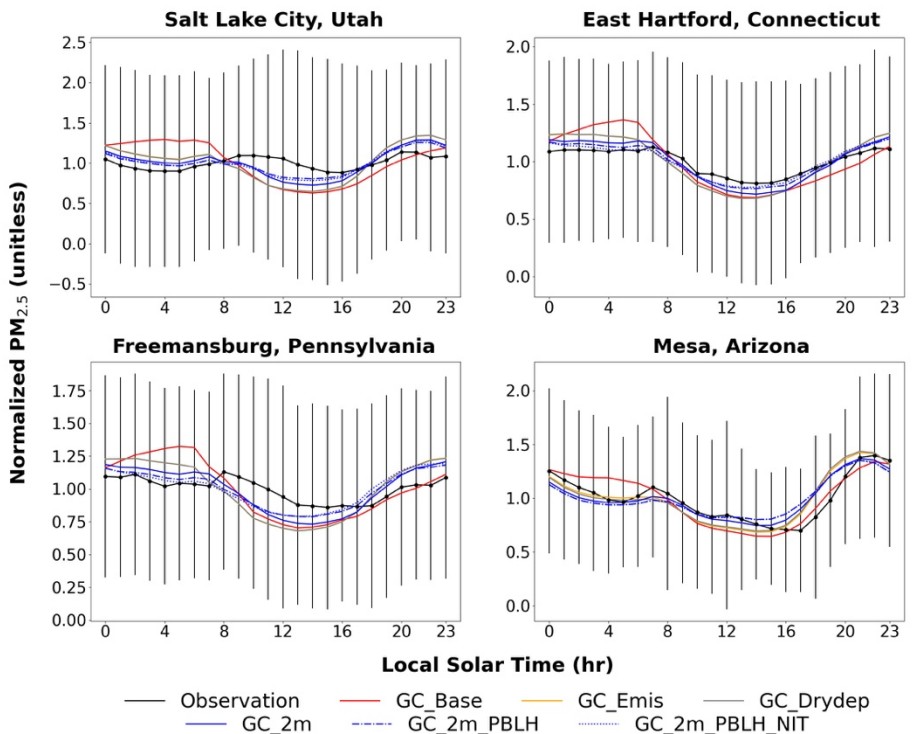

**Figure 10. Diel PM$_{2.5}$ mass variation in the GEOS-Chem simulations (Table 1) and the in situ measurements over four**
**example FEM sites.**

Despite the pronounced improvement in simulating the PM$_{2.5}$ diel variation, positive biases remain in early morning
in most regions and seasons (Fig. 9). The regional and seasonal variation in PM$_{2.5}$ chemical composition offers insight
(Fig. S11). Nitrate appears to be an important contributor to the bias, which is not fully understood as discussed in
Sect. 5.6. Insufficient vertical and horizontal resolution in our simulations to fully resolve nocturnal stratification and
horizontal source separation (Zakoura and Pandis, 2018; Boys 2022) are possible contributors. The remaining evening
bias in the Spring, Summer, and Winter in the central US could reflect the possible underestimation of residential
emissions in NEI (Trojanowski et al., 2022). Fig. S4 shows that the OC emissions as a relevant indicator of residential
combustion are the weakest in the evenings for spring, summer, winter in Central US.

In summary, emissions, vertical representativeness differences between model and observations, boundary layer
mixing, and nitrate are found to be the top four contributing factors of the diel biases in GEOS-Chem PM$_{2.5}$. Dry
deposition and scavenging by the formation of dew are relatively unimportant. The vertical correction for the
representativeness differences by using the resistance-in-series method is critical for improving the simulation of the
PM$_{2.5}$ diel amplitude as well as capturing the timings of the observed PM$_{2.5}$ morning peak and afternoon minimum,

indicating the significance of vertical resolution of GEOS-Chem for simulating diel $PM_{2.5}$ variation. Reducing the daytime positive biases in GEOS-FP PBLH and improvements of the diel representation of residential combustion may be useful to further improve the diel $PM_{2.5}$ in GEOS-Chem. In addition to the above impacting factors, we emphasize the necessity of conducting simulations at fine spatial resolution to resolve processes affecting diel variation of $PM_{2.5}$ concentrations. Comparison of the GEOS-Chem simulations at $0.25° \times 0.3125°$ and $2° \times 2.5°$ against the

FEM observations (Fig. S12) reveals that higher spatial resolution better enables the model to reproduce the observed diel $PM_{2.5}$ variation through reducing the excessive $PM_{2.5}$ accumulation during nighttime (18:00 LT- 6:00 LT). At the coarse spatial resolution, the simulated $PM_{2.5}$ mass concentrations increase by 36.3% from 18:00 LT to 6:00 LT, greater than the observed 5.8% increase. At the finer spatial resolution, that nighttime increase of $PM_{2.5}$ mass concentrations reduces to 20.3%. The recent advances to the High-Performance implementation of the GEOS-Chem

model (GCHP) Model with stretched grid capabilities (Bindle et al., 2021; Martin et al., 2022) enables higher spatial resolution than $0.25° \times 0.3125°$, which could offer improved representation of resolution-dependent processes in future analyses.

## 7 Conclusions

In this work, we used the GEOS-Chem model in its nested configuration to interpret the observed diel variation in

$PM_{2.5}$ concentration for the contiguous United States. We identified and addressed several biases of the base GEOS-Chem simulation of the diel variation of $PM_{2.5}$ mass concentrations. 1) The simulated $PM_{2.5}$ accumulation overnight was excessive in the base simulation, which disagreed with the observed concentration decrease or flatness from midnight to early morning, leading to a significantly overestimated $PM_{2.5}$ diel amplitude in the model. 2) The simulated timings of the $PM_{2.5}$ morning peak and afternoon minima were notably earlier relative to the in situ observations,

especially for the morning peak (3 hours earlier).

To reveal the contributing factors to the diel $PM_{2.5}$ biases in the base simulation, we conduct sensitivity simulations in which we 1) increased the temporal resolution of anthropogenic emissions from monthly to hourly, 2) updated the dry deposition scheme, 3) resolved the vertical representativeness differences between the model and the observations, 4) corrected for the diel biases in the boundary layer heights of the model, 5) explored the impacts from dew formation

and 6) examined the role of aerosol nitrate.

We found that several developments aided representation of the $PM_{2.5}$ diel variation in the GEOS-Chem model. Hourly representation of emissions decreased normalized $PM_{2.5}$ concentrations at night with increases during the day. Accounting for vertical representativeness differences between the GEOS-Chem surface layer of 120m and the measurement height of 2m further decreases $PM_{2.5}$ at night, leading to better representation of the timing of the

morning peak (~7am) and afternoon minimum. Developments to the dry deposition scheme aided mechanistic representation of gravitational settling and its hygroscopic dependence, albeit with negligible effects on $PM_{2.5}$ diel variation. Reduction of simulated PBLH to represent aircraft observations also aids agreement with observed $PM_{2.5}$ diel variation. These improvements also partially addressed a longstanding issue of a positive bias in simulated nitrate concentrations but additional constraints from nitrate observations were necessary to represent diel $PM_{2.5}$ variation.

The slight $PM_{2.5}$ decrease/flatness overnight is more likely caused by diminished emissions, rather than enhanced dry deposition (Zhao et al., 2009) or dew events (Sect. 5.5). Hourly anthropogenic emissions are important for GEOS-Chem to accurately simulate diel $PM_{2.5}$ variation. Using monthly emissions combined with sector or species-specific diel scaling factors instead can lead to higher $PM_{2.5}$ positive biases overnight. Resolving the vertical representativeness differences introduced by subgrid vertical gradient of $PM_{2.5}$ in the surface model level contributed to capturing timings

of $PM_{2.5}$ diel variation. Overall, the mean diel variation in $PM_{2.5}$ for the US is attributed to 1) growth in $PM_{2.5}$ concentrations by 10% from early morning (4:00 LT) to mid-morning (8:00 LT) driven by increasing emissions into a shallow mixed layer, 2) subsequent decline in $PM_{2.5}$ concentrations by 22% from mid-morning (8:00 LT) to late afternoon (15:00 LT) during growth of the mixed layer, 3) rapid increase in $PM_{2.5}$ by 30% from late afternoon (15:00 LT) to evening (22:00 LT) as emissions persist into a collapsing mixed layer, and 4) subsequent weak decline in $PM_{2.5}$

concentrations by 10% as emissions diminish overnight (22:00 LT – 4:00 LT). Despite the advances in representing and understanding $PM_{2.5}$ diel variation, minor biases remain. A more mechanistic representation of nitrate is needed. The importance of vertical resolution in representing $PM_{2.5}$ diel variation identifies an advantage to be offered by a forthcoming GEOS-6 dataset with a planned doubled number of vertical levels in the PBL compared to GEOS-FP (NASA, 2012). Recent advances in the horizontal resolution of GEOS-Chem (Bindle et al., 2021; Martin et al., 2022)

should also enable simulations with finer spatial resolution to further improve the diel performances.





*Code/Data availability.* The hourly FEM and 24-hour average FRM $PM_{2.5}$ in situ measurements are available at https://aqs.epa.gov/aqsweb/airdata/download_files.html. The hourly RH measurements at four example sites in Fig. 6 are available at https://www.ncei.noaa.gov/maps/lcd/. The AMDAR PBLH data is available at https://zenodo.org/record/3934378#.YiExLZZOk2y.

*Author contributions.* YL and RVM designed the study. YL performed the model simulations and the data analysis. CL and AVD contributed to the diel analysis of $PM_{2.5}$. BLB contributed to the model developments of aerosol dry deposition and the correction on $PM_{2.5}$ vertical representativeness. JM contributed to preparing emission data for the simulations. JRP contributed to the investigation on the impacts of PBLH on diel $PM_{2.5}$.

*Competing interests.* None of the authors has any competing interests.

*Financial support.* This work was supported by NASA Grants 80NSSC21K0508 and 80NSSC21K0429.

*Acknowledgements.* Thanks to Ethan W. Emerson and Delphine K. Farmer for constructive comments about the science of aerosol dry deposition. We thank Barron H. Henderson for making available sectoral diel scaling factors for the CEDS inventory.

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
