# Peer review of "Development and evaluation of processes affecting simulation of diel fine particulate matter variation in the GEOS-Chem model"

_EGUsphere, 2023_

## Referee Comment (RC2)

Review of Li et al., 2023

The authors assess the ability of the GEOS-Chem chemical transport model to simulate the average diel variability in $PM_{2.5}$ over the United States in 2016. They perform sensitivity simulations to test improvements to the representation of anthropogenic emissions, dry deposition, boundary layer height, and vertical representation errors. Overall, the authors find that hourly anthropogenic emissions and correcting from the lowest model layer midpoint to the measurement altitude significantly improve the diel variation of simulated $PM_{2.5}$.

**Major Comments**

Overall, the authors should better discuss the reasons why nitrate aerosol drives the diurnal behavior in the model, and what chemical pathways produce this nitrate in GEOS-Chem. The authors do split their analysis into season and region. However, it could provide much more insight into $PM_{2.5}$ diel behavior and model issues if the authors examined rural vs. urban behavior. With these revisions, this paper would be appropriate for publication in ACP.

**Specific Comments**

Line 63 – Which SOA scheme is being used? Simple, complex, VBS? Could the authors break down SOA and POA in their speciated $PM_{2.5}$ plots?
Line 80 – NEI 2016 is available in GEOS-Chem and appears to be default according to the website. Why was this not used? http://wiki.seas.harvard.edu/geos-chem/index.php/EPA/NEI11_North_American_emissions
Line 84 - Are any anthropogenic or fire emissions emitted above the first model level?
Line 100 – The authors could highlight upfront that there has been a lot of recent developments in understanding particulate deposition and clarify for the reader the need to investigate how important these developments could be for understanding $PM_{2.5}$ variability.
Section 3 – At what RH do these methods measure? Is the model $PM_{2.5}$ adjusted for this?
Line 136 – It looks like the first peak is at 8am, that is not really 'mid-morning'.
Line 142 – The authors should discuss which components of $PM_{2.5}$ are driving the diurnal cycle. For example, recent work has described model nighttime overestimates in nitrate (Miao et al., 2020; Travis et al., 2022; Zhai et al., 2021). It might be better to describe the components driving the diurnal cycle behavior directly after Fig. 2 rather than later on.
Figure 2 – It would be very informative if Fig. 2 was split into urban and rural as there are likely large difference in dynamics, emissions, chemistry etc. between them.
Line 155 – It is very difficult to think about how emissions drive errors in $PM_{2.5}$ without first seeing the speciated $PM_{2.5}$ diurnal cycle.
Line 250 – The authors might consider that if the model nitrate is overestimated, hygroscopic growth might also be too larger.
Line 261 – Please provide references for this statement. "Vertical concentration gradients within 120 m of the surface have been widely observed for aerosol species in previous field campaigns."
Line 265 – How different is the temperature and RH between 2m and the lowest model level, and how would that impact thermodynamic partitioning of sulfate-nitrate-ammonium aerosol?
Line 306 – Please briefly describe how the AMDAR PBLH is generated.
Line 313 – Please comment more on why the PBLH in GEOS-FP might be too high and what that could mean for other parts of the simulation (e.g., temperature).
Line 315 – Could you explain why it would be reasonable to have a PBLH maximum in spring?
Line 384 – See the additional two papers on nitrate overestimates (above Line 142). It would be worth commenting on the reasons described in those papers. The authors should comment that a large bias remains in their modeled nitrate. For example, there seems to be general consensus that $NO_2$ hydrolysis shouldn't be very important in the model and that $N_2O_5$ hydrolysis should be the main pathway (one example https://agupubs.onlinelibrary.wiley.com/doi/full/10.1029/2019GL085498). Which pathways dominate the authors' results? The authors could consider turning $NO_2$ hydrolysis off if it seems unreasonably large.
Line 425 – Could you be quantitative about the impact of resolution?
Line 456 – What is this product and what will the increased resolution be in the PBL?

---

## Author Comment (AC1)

Thanks to the reviewers for their valuable comments and suggestions, which are helpful to improve this manuscript. Our responses (blue) are detailed as below, in which the figure, section and line numbers correspond to the revised manuscript.

Reviewer 1

Li et al. present a comprehensive evaluation of various processes affecting the simulation of diel variation of $PM_{2.5}$ in the United States in 2016 using the GEOS-Chem in a chemical transport model (CTM), 0.25-degree configuration. The base, unmodified model presented a 105% high bias compared to observations; the authors investigate the effects of temporal resolution, hourly vs. monthly averaged emission inventory temporal resolution, resolution of vertical gradient in the lowest model level, revised dry deposition parameterization, and adjustments to boundary layer height (PBLH) on improving this bias. The work is a useful reference for the effects of these factors and well fit for publication in Atmospheric Chemistry and Physics. I have minor comments prior to recommending the manuscript for publication.

Thanks for the positive comments on the manuscript. Point-by-point responses are provided as below.

Major comments:

1. The authors use the term "vertical representativeness" in the abstract (L8) and later on in the text to represent the correction of $PM_{2.5}$ masses from model level center (which is the conventional way we interpret mass in a model vertical level) to the height of surface measurements, correcting for aerodynamic resistance. I understand this is a complex concept to explain but I wished that it could be explained first in the abstract then defined as the term "vertical representativeness". Or maybe clarify this as the impact of modeling the subgrid vertical gradient (this would be more specific and easier to understand). This would make the text easier to understand.

Thanks for the suggestion. In the revised manuscript, we used "subgrid vertical gradient" instead in the abstract so that readers can understand more easily. We formally defined and explained the "vertical representativeness" in the introduction to avoid any confusions afterward. Detailed revisions are listed as below.

Line 7: "We find that temporal resolution of emissions, subgrid vertical gradient between model level center and observations, as well as boundary layer mixing are the major causes for this inconsistency."

Line 48: "As subgrid vertical gradients exist between model level center and surface observations, CTM simulation and in-situ measurement stand for $PM_{2.5}$ at different altitudes. This is the so-called vertical representativeness difference, which can affect model evaluation."

2. The abstract and conclusion say that the $PM_{2.5}$ diel variation is "driven by ... 1) to 4)". Perhaps I am missing something here, but these conclusions kind of reflect what we already know about pollutants and their interaction with boundary layer dynamics, perhaps a little too close to the textbook. Could the authors, given the specific conclusions about modeling processes affecting diel $PM_{2.5}$ variation in their work, elaborate and provide more insight on model representation that could be derived from this work, and how we could improve diel $PM_{2.5}$ variation simulation in general in models or GEOS-Chem in particular?

Thanks. We added more specific conclusions with more insights. In the abstract and conclusion of the revised manuscript, we emphasized the importance of spatially resolved diel variation in anthropogenic emissions ("true hourly") and resolving the vertical representative differences introduced by subgrid vertical gradient on improving the simulation of diel $PM_{2.5}$ variation in GEOS-Chem, instead of describing driving forces generally. We also specified the magnitude of the changes and the hours associated with each change. Detailed revisions are listed as below.

Line 13: "Gridded hourly emissions rather than diel scaling factors applied to monthly emissions reduced biases in simulated $PM_{2.5}$ overnight. Resolving the subgrid vertical gradient in the surface model level aided capturing timings of $PM_{2.5}$ morning peak and afternoon minimum."

Line 15: "Based on the improved model, we find that the mean observed diel variation in $PM_{2.5}$ for the US is driven by 1) building up of $PM_{2.5}$ by 10% in early morning (4:00 - 8:00 local time, LT) due to increasing anthropogenic emissions into a shallow mixed layer, 2) decreasing $PM_{2.5}$ by 22% from mid-morning (8:00 LT) through afternoon (15:00 LT) associated with mixed layer growth, 3) increasing $PM_{2.5}$ by 30% from mid-afternoon (15:00 LT) though evening (22:00 LT) as emissions persist into a collapsing mixed layer, and 4) decreasing $PM_{2.5}$ by 10% overnight (22:00 - 4:00 LT) as emissions diminish."

Line 550: "The slight $PM_{2.5}$ decrease/flatness overnight is more likely caused by diminished emissions, rather than enhanced dry deposition (Zhao et al., 2009) or dew events (Sect. 5.5). Hourly anthropogenic emissions are important for GEOS-Chem to accurately simulate diel $PM_{2.5}$ variation. Using monthly emissions combined with sector or species-specific diel scaling factors instead can lead to higher $PM_{2.5}$ positive biases overnight. Resolving the vertical representativeness differences introduced by subgrid vertical gradient of $PM_{2.5}$ in the surface model level contributed to capturing timings of $PM_{2.5}$ diel variation. "

3. The use of hourly temporal resolution inventory vs. a monthly mean inventory in NEI presents interesting implications. The authors mention that monthly mean inventories are usually all we get, and that's true for most of the world. Does GEOS-Chem / HEMCO not have an anthropogenic diurnal profile for emissions applied over the monthly mean data? It would be strange not to.

If HEMCO applies a diurnal profile to the monthly mean data in NEI by default, then the work here is considering the impact of a more accurate diurnal profile from the actual inventory hourly data versus the default "prescribed" profile that is applied constantly, and it would be quite surprising to find such an improvement in the diel amplitide bias by simply improving the diurnal profile.

If HEMCO does not apply such profile, it seems to me it is an obvious oversight in the model. What if the normalized profile shown in Figure 3 was applied to every day in the simulation? How much improvement would it yield, and how much compared to a "true" hourly emissions input? There are other implications here if simply applying the normalized profile could get us most of the benefits, because reading hourly data is computationally expensive, especially as we move to higher resolution.

Thanks for these valuable comments and questions which help us to improve the analysis on the impacts of emissions on simulated diel $PM_{2.5}$ in GEOS-Chem in the manuscript.

GEOS-Chem/HEMCO does have default diel profiles which are species-specific. The diel profiles for nitrogen oxides ($NO_x$) are spatially resolved. Non-methane volatile organic compounds (NMVOCs),

carbon monoxide (CO), ammonia ($NH_3$), sulfur dioxide ($SO_2$), black carbon (BC) and organic carbon (OC) share another spatially uniform diel profile (Fig. S5a).

In Table 1, GC_Emis uses the "true" hourly NEI emissions and GC_Base uses the monthly NEI emissions with the default diel scaling factors (Fig. S5a) not applied. We now also conduct comparisons between the simulation using the "true" hourly emissions and the simulation combining monthly emissions with diel scaling factors.

In the revised manuscript, we add three extra sensitivity simulations as described in Table S2. GC_Emis_S1 uses monthly NEI emissions with the default diel scaling factors (Fig. S5a) in GEOS-Chem v12.6.0. GC_Emis_S2 uses monthly NEI emissions with the averaged profile shown in Figure 4 in the revised manuscript as suggested by the reviewer. GC_Emis_S3 uses monthly CEDS emissions with sectoral diel scaling factors (Fig. S5b), which is proposed as the default emission configuration of global simulations in GEOS-Chem 14 (https://github.com/geoschem/geos-chem/issues/1824). Results (Fig. S6) show that the diel performance of GC_Emis outperforms all sensitivity simulations in Table S2 when comparing with FEM measurements, indicating heterogeneities of diel variation of anthropogenic emissions in different place and date as well as the significance of using "true" hourly emissions in simulating diel $PM_{2.5}$ in GEOS-Chem.

Specific updates are listed below.

Table S2, Fig. S5, and Fig. S6 were added to the supplement. Table S2 describes the design of the supplementary simulations. Fig. S5 shows the default diel scaling factors used in GEOS-Chem v12.6.0 and the proposed diel scaling factors used in proposed for GEOS-Chem 14. Fig. S6 shows the diel $PM_{2.5}$ results of these supplementary simulations and GC_Emis which uses the "true" hourly NEI emissions.

Line 13: "Gridded hourly emissions rather than diel scaling factors applied to monthly emissions reduced biases in simulated $PM_{2.5}$ overnight."

Line 41: "Global anthropogenic emission inventories are generally available at monthly mean resolution (Janssens-Maenhout et al., 2015; Huang et al., 2017; McDuffie et al., 2020). These monthly inventories are often applied as is for a wide range of studies. Some national emission inventories (e.g., NEI) contain local species- and sector-specific diel variation. Such national information for a specific country has in some instances been applied to provide diel information for global inventories in some models. There is need to explore the effects of these different approaches upon the diel variation in $PM_{2.5}$ concentrations."

Line 240: "However, over most regions worldwide, only monthly emissions are available with crude diel scaling factors from specific regions as a possible proxy for hourly emissions. To assess the performance of such diel scalars in simulating diel $PM_{2.5}$, we conducted three supplementary sensitivity simulations in Table S2, in which sector or species-specific diel scaling factors (Fig. S5) are applied to NEI and CEDS monthly emissions. Results (Fig. S6) show that the average $PM_{2.5}$ accumulation overnight (0:00 LT – 6:00 LT) among the supplementary cases is 2.6 times of that in GC_Emis, leading to stronger overestimation of $PM_{2.5}$ overnight. To optimize the model performance in simulating diel $PM_{2.5}$, hourly gridded emissions are preferred over using monthly emissions with scaling factors. Nevertheless, the diel emission profile does not fully explain the diel biases identified in Sect. 4. Other contributing factors exist."

Line 551: " Hourly anthropogenic emissions are important for GEOS-Chem to accurately simulate diel $PM_{2.5}$ variation. Using monthly emissions combined with sector or species-specific diel scaling factors instead can lead to higher $PM_{2.5}$ positive biases overnight."

Specific (minor) comments:

1. L31: WRF-Chem is not a CTM (which usually implies offline meteorology), it is driven by online meteorology from WRF. Also it would be useful to state the mechanism used in WRF-Chem here as it has a wide range of configurations and it helps to be specific.

Thanks. Revised.

Line 34: " Du et al. (2020) used the WRF-Chem model (Grell et al., 2005) with the MOSAIC (Model for Simulating Aerosol Interactions and Chemistry) scheme and the CBM-Z (carbon bond mechanism) photochemical mechanism to simulate diel $PM_{2.5}$ variation over East Asia and found nighttime overestimation, possibly due to insufficient boundary layer mixing. "

2. L36: Another minor comment, but I suggest "lowest model level" instead of "first model level". First level can be ambiguous; some models (such as CESM or GEOS-5) have first level as top of atmosphere.

Thanks. Revised.

Line 47: "The vertical extent of the lowest model level in CTMs is typically tens of meters above ground, while ground-based measurements are taken at around two meters. "

3. L56: Was GEOS-FP or MERRA2 used in this work? This has some implications as the PBLH used could be different. The authors don't include the PBLH in the final result and only point to its potential importance, but I think that it is fine. Fixing "PBLH" is only a band-aid because PBLH is a diagnostic from the GEOS output; the PBL mixing scheme in GEOS-Chem takes that PBLH diagnostic value to do the mixing, but it would be inconsistent with GEOS dynamics. But improvements to the PBLH value itself passed to GEOS-Chem can provide improvements to model PM2.5 simulation (e.g., as demonstrated by improved PBLH driving improved PM2.5 in the WRF-GC model, which uses the GEOS-Chem PBL mixing routines as well) and this work further confirms this conclusion.

GEOS-FP was used in this work as mentioned at Line 75 in the original manuscript. To make this clearer, in the revised manuscript, we first stated that our model is driven by GEOS-FP at the start of Sect. 2.1. Line 81: "We use the GEOS-Chem chemical transport model version 12.6.0 (www.geos-chem.org) driven by the GEOS-5 Forward Processing (GEOS-FP) assimilated meteorology from the NASA Global Modeling and Assimilation Office (GMAO) to examine the factors controlling the diel $PM_{2.5}$ mass variations." Then the variables in GEOS-FP are further explained later at Line 94: "GEOS-FP is used for meteorological inputs, which includes hourly surface variables and 3-D variables at every 3 hours."

Thanks for the positive comments on our developments of the PBLH constraint to improve diel $PM_{2.5}$ in GEOS-Chem and good to learn that the results are consistent with the WRF-GC model. To address comments from Reviewer 1, we included the PBLH constraint in the final result as suggested in the revised manuscript. Relevant contents were updated accordingly as detailed below.

Line 9: "We applied an hourly anthropogenic emission inventory, converted the $PM_{2.5}$ mass concentrations from model level center to the height of surface measurements by correcting for aerodynamic resistance, adjusted the boundary layer heights in the driving meteorological fields using aircraft observations, and constrained nitrate concentrations using in situ measurements. The bias in the $PM_{2.5}$ diel amplitude was reduced to -12% in the improved simulation."

Figures 3, 8, S2, S8, and S11 updated to show results in GC_2m_PBLH and GC_2m_PBLH_NIT.

Line 450: "Overall updating the temporal resolution of emissions, dry deposition parameterizations, boundary layer height, resolving the vertical representative differences between model and observations and constraining nitrate notably improves the diel variation of $PM_{2.5}$ in GC_2m_PBLH_NIT relative to GC_Base for both urban and rural regions (Fig. S3) in a similar way. In the annual diel comparison averaged across the US (Fig. 2), the bias in the $PM_{2.5}$ diel amplitude in GC_2m_PBLH_NIT (-12%) is significantly reduced relative to GC_Base (106%)."

Line 455: "GC_Base simulates them with biases of -3 and -1 hours while GC_2m_PBLH_NIT agrees with observed timing within 1 hour."

Line 459: "By correcting for the vertical representativeness differences, using emissions with hourly temporal resolution, adjusting the GEOS-FP boundary layer heights, and constraining nitrate concentrations, these biases are largely addressed in GC_2m_PBLH_NIT with the distribution in the histogram better matching observations. The RMSD of diel $PM_{2.5}$ between GC_2m_PBLH_NIT and the FEM observations decreases from 2.18 to 0.75 $\mu g/m^3$."

Line 463: "With reduced the 24-hour averaged $PM_{2.5}$ concentration, GC_2m_PBLH_NIT also improves the agreements of annual-mean $PM_{2.5}$ against the FEM/FRM measurements across the US (Sect. S2)."

Line 472: "The GC_2m_PBLH_NIT simulation generally reproduces this summer minimum in diel amplitude, improving on GC_Base which simulates the minimum amplitude in winter, by reducing excess $PM_{2.5}$ at night, by reducing $PM_{2.5}$ precursor emissions and by accounting for vertical representativeness differences at night, by adjusting boundary layer height using aircraft observations and by constraining nitrate."

Line 493: "By further correcting for the vertical representativeness differences, adjusting boundary layer height and constraining nitrate, in GC_2m, GC_2m_PBLH and GC_2m_PBLH_NIT, the simulations more closely represents the FEM measurements."

Sect. S2: "... … Fig. S2b maps the annual concentrations by the GC_2m_PBLH_NIT simulation, in which temporal resolution of emissions is increased from monthly to hourly, dry deposition scheme is updated, boundary layer height is adjusted, the vertical representativeness differences between model and observations are resolved and nitrate is constrained. The RMSD of the GC_2m_PBLH_NIT $PM_{2.5}$ against the FEM/FRM measurements drops from 3.35/3.75 to 2.74/2.84 $\mu g/m^3$. The overestimation of $PM_{2.5}$ in Eastern US and the west coast is reduced. These results indicate that our model updates improve on the simulation of annual mean concentrations."

4. L75 / Table 1 & L150 / Intro of Section 5: I suggest adjusting the table columns to use similar terminology and the same order as they're mentioned in the text.

Thanks. Revised in Table 1.

5. L134: To confirm, the observations in one GEOS-Chem 0.25x0.3125 grid box are averaged for the purpose of comparing to the model, or the closest site to grid box center are used?

The closest site to grid box center is used. We added the corresponding description of matching GEOS-Chem grid and in situ measurements at the end of Sect. 3. At line 156: "To compare with GEOS-Chem, each site is matched with the GEOS-Chem grid nearest box center."

6. Figure 4: Please use consistent unit labeling (cm s-1 in (a) and cm/s in (b)). "constrains" -> "constraints" in the figure legend.

Thanks. Corrected.

7. L252-L254: It's good to know that diel PM2.5 variation is shown to be insensitive to updates in dry deposition parameterization. Out of curiosity, have the dry deposition updates affected other aspects of the GEOS-Chem simulation or certain aerosol species in particular?

The update on surface resistances in dry deposition has been shown to strongly affect number concentrations of modeled aerosol with diameter between 100 and 500 nm at the surface and the cloud-droplet number concentrations (CDNC) at low-cloud level (~900 hPa) in the GEOS-Chem TOMAS (TwO-Moment Aerosol Sectional) model (Emerson et al., 2020).

8. L423: The authors briefly mention the effect of horizontal resolution in improved representation of PM2.5 diel amplitude and timings of the min/maxima - this is consistent with recent model sensitivity experiments (e.g., in MUSICA/CAM-chem) where improved horizontal resolution improves model representation despite the same underlying physics/dynamics/chemistry. Considering the authors' development of improved representation of vertical gradient in the lowest model layer, do the authors also think that improved vertical resolution in the lowest model levels would help as well, instead of applying corrections?

Interesting question. We do think that improved vertical resolution in the lowest model levels could help. The vertical correction on aerodynamic resistance used in this work assumes dry deposition dominates the processes affecting the subgrid vertical gradient between the lowest model level center and surface measurements, which is a simplified approach without doing realistic chemistry transport modeling. Using the lowest model level closer to the ground in future is possibly a better solution if the thinner surface layer can be precisely represented. We comment on this in the final paragraph (Line 562: "The importance of vertical resolution in representing $PM_{2.5}$ diel variation identifies an advantage to be offered by a forthcoming GEOS-6 dataset with a planned doubled number of vertical levels in the PBL compared to GEOS-FP (NASA, 2012).").

Reviewer 2

Overall, the authors should better discuss the reasons why nitrate aerosol drives the diurnal behavior in the model, and what chemical pathways produce this nitrate in GEOS-Chem. The authors do split their analysis into season and region. However, it could provide much more insight into $PM_{2.5}$ diel behavior

and model issues if the authors examined rural vs. urban behavior. With these revisions, this paper would be appropriate for publication in ACP.

Thanks for these valuable suggestions and comments. In the revised manuscript, we identified $N_2O_5$ hydrolysis as the dominant pathway of nighttime nitrate formation in our simulations by conducting sensitivity tests turning off different mechanisms. By referencing the analysis by Miao et al., (2020) and Travis et al., (2022) and co-analysis with the results in this work, we further discuss the driving factors of nighttime nitrate biases and the impacts on simulated diel $PM_{2.5}$. In addition, the diel analyses were separated for urban and rural based on gridded urban extent data. Point-to-point responses are listed below.

**Specific Comments**

Line 63 – Which SOA scheme is being used? Simple, complex, VBS? Could the authors break down SOA and POA in their speciated $PM_{2.5}$ plots?

Simple SOA is used. To make this clearer, in the revised manuscript at Line 90, we stated that: "The so-called "simple" scheme (Kim et al., 2015) is used for simulating secondary organic aerosol (SOA)."Yes, the revised composition plots contain both SOA and POA in Fig. 3 and Fig. S10, Fig. S11. Full names of POA, SOA and BC are also added in the caption of Fig. 3.

Line 80 – NEI 2016 is available in GEOS-Chem and appears to be default according to the website. Why was this not used? http://wiki.seas.harvard.edu/geos-chem/index.php/EPA/NEI11_North_American_emissions.

The NEI 2016 inventory developed in the GEOS-Chem community (http://geoschemdata.wustl.edu/ExtData/HEMCO/NEI2016/v2021-06/) has only monthly resolution. To investigate the impact of temporal resolution of emissions on diel $PM_{2.5}$, requires the NEI 2011 with both monthly and hourly emissions. Yearly scaling factors were applied to scale NEI 2011 to the target year of 2016. We add Line 112 in the revised manuscript: "We do not use the NEI 2016 inventory since that inventory is only available at monthly resolution in GEOS-Chem."

Line 84 - Are any anthropogenic or fire emissions emitted above the first model level?

Yes. All point sources in NEI are three-dimensional, including electric generating units (EGUs), peaking electric generating units, oil & gas, non-EGU industrial stacks, which can emit above the first model level. Taking EGU $SO_2$ emission as an example, in January of 2016, 65.5% of the emissions are emitted in the 2nd model level, 32.5% of the emissions are emitted in the 3rd model level and only 2.0% of the emissions are emitted in the 1st model level.

To make this clearer, we stated in the revised manuscript that NEI point sources are vertically resolved at Line 110: "Point sources in the NEI inventory are all vertically resolved, which mainly include large industrial facilities, power plants and airports."

Line 100 – The authors could highlight upfront that there has been a lot of recent developments in understanding particulate deposition and clarify for the reader the need to investigate how important these developments could be for understanding $PM_{2.5}$ variability.

Thanks for the suggestion. In the revised manuscript, we added a paragraph to highlight recent developments of particle dry deposition scheme and stated that their impacts on diel $PM_{2.5}$ variation are unclear at Line 54:

"Aerosol dry deposition, defined as the removal of aerosols by gravitational settling, by Brownian diffusion, or by impaction and interception resulting from turbulent transfer (Beckett et al., 1998), is an important sink process. Recent investigations have examined developments to the dry deposition scheme used in CTMs. Petroff and Zhang (2010) developed a sized-resolved particle dry deposition scheme with a new surface resistance parameterization by simplification of a one-dimensional aerosol transport model. Kouznetsov and Sofiev (2012) proposed a comprehensive particle dry deposition scheme which accounts for physical properties of the air flow, surface and depositing particles. Zhang and Shao (2014) improved the modeling of particle dry deposition on rough surfaces by treating gravitational settling analytically and considering the roughness in particle diffusion and surface collection. Emerson et al. (2020) revised size-resolved particle dry deposition through constraining the surface resistances using particle flux observations. The impacts of recent updates on $PM_{2.5}$ mass concentrations and its diel variation remains unclear."

Section 3 – At what RH do these methods measure? Is the model $PM_{2.5}$ adjusted for this?

In Section 3, the FEM measurements we used in the diel analysis are provided from four types of instruments, which are Met One BAM-1020 Mass Monitor, Thermo Scientific 5014i, Thermo Scientific TEOM 1405-DF and Thermo Scientific Model 5030 SHARP. As stated in the EPA List of Designated Reference and Equivalent Methods (EPA, 2023), BAM-1020 and 5014i measure at 35% RH. As described in the Instruction Manual of Thermo Scientific Model 5030 SHARP (Thermo Fisher Scientific, 2013), the SHARP 5030 also measures $PM_{2.5}$ at 35% RH. For the TEOM 1405-DF, the EPA Standard Operating Procedure for the continuous Measurement of Particulate Matter (EPA, 2021) suggests measuring $PM_{2.5}$ under 30-35% RH to reduce the interference from aerosol water. The FRM $PM_{2.5}$ measurements are also generally well controlled at around 35% RH (EPA, 2007). We describe these in the revised manuscript at line 156: "To compare with GEOS-Chem, each site is matched with the GEOS-Chem grid nearest box center. The FRM and FEM measurements used in this work are at 35±5% relative humidity (EPA, 2007; Thermo Fisher Scientific, 2013; EPA, 2021; EPA, 2023)."

Yes, based on the above information, we calculate $PM_{2.5}$ masses considering hygroscopic growth at 35% RH following the guideline provided by the Aerosol Working Group in the GEOS-Chem community (GEOS-Chem Aerosols Working Group, 2021).

To make this clearer, we now state in Section 3 at line 159: "To match the measurement RH, the GEOS-Chem $PM_{2.5}$ and its composition were calculated considering the corresponding hygroscopic growth following standard practice in GEOS-Chem (GEOS-Chem Aerosols Working Group, 2021)."

Line 136 – It looks like the first peak is at 8am, that is not really 'mid-morning'.

Thanks. Revised to 8am at Line 170.

Line 142 – The authors should discuss which components of $PM_{2.5}$ are driving the diurnal cycle. For example, recent work has described model nighttime overestimates in nitrate (Miao et al., 2020; Travis et al., 2022; Zhai et al., 2021). It might be better to describe the components driving the diurnal cycle behavior directly after Fig. 2 rather than later on.

Thanks for the suggestion. Done. Now at Line 191-199.

Figure 2 – It would be very informative if Fig. 2 was split into urban and rural as there are likely large difference in dynamics, emissions, chemistry etc. between them.

Added the plot of urban and rural diel $PM_{2.5}$ to the supplement as Fig. S3.

Added at Line 180: "We classify each FEM measurement and the corresponding GEOS-Chem simulation into urban and rural using the Global Rural-Urban Mapping Project (GRUMP) v1 (Balk et al., 2006) data at 30 seconds resolution. Results (Fig. S3) indicate that the observed diel variations of $PM_{2.5}$ in urban and rural areas across the US are highly consistent (r=0.97). Both urban and rural sites exhibit the same bi-modal patterns with $PM_{2.5}$ peaks near 8:00 LT and 21:00 LT, and minima near 4:00 LT and 16:00 LT. The $PM_{2.5}$ dips near 4:00 LT and 16:00 LT are deeper over urban regions than over rural regions, which may reflect stronger vertical mixing from the urban heat island effect (Travis et al., 2022). The consistency of diel $PM_{2.5}$ variation across urban and rural locations implies a dominant role from natural processes."

Added at Line 450: "Overall updating the temporal resolution of emissions, dry deposition parameterizations, boundary layer height, resolving the vertical representative differences between model and observations and constraining nitrate notably improves the diel variation of $PM_{2.5}$ in GC_2m_PBLH_NIT relative to GC_Base for both urban and rural regions (Fig. S3) in a similar way."

Line 155 – It is very difficult to think about how emissions drive errors in $PM_{2.5}$ without first seeing the speciated $PM_{2.5}$ diurnal cycle.

Thanks for the suggestion. In the revised manuscript, the speciated $PM_{2.5}$ diel variation is provided in Fig. 3 before the discussion of impacts of emissions. We also added discussion on the impacts of emissions on the diel variation of $PM_{2.5}$ components at Line 229: " In terms of composition (Fig. 3b), the average mass concentrations of BC and POA overnight (0:00 LT - 6:00 LT) decrease by 25.7% and 12.9%, contributing the most to the reduced overnight $PM_{2.5}$ accumulation. Sulfate concentrations overnight decrease by only 3.5% due to weak day-night contrast in $SO_2$ emissions. Nitrate and ammonium concentrations decrease by only 7.1% and 6.3%, reflecting the relatively minor role of primary emissions versus secondary production for these two species. In GC_Emis, nitrate still accumulates notably (by 23.1%) from 0:00 LT to 6:00 LT, acting as the major contributor of the $PM_{2.5}$ nighttime bias."

Line 250 – The authors might consider that if the model nitrate is overestimated, hygroscopic growth might also be too larger.

Thanks for the idea. In the revised manuscript, we separate out aerosol water in all compositional figures (Fig. 3, Fig. S10, Fig. S11) from dry masses of aerosol components. We also developed the GC_2m_PBLH_NIT simulation, in which nitrate and its associated water are reduced to match observations following the slope of reduced major axis regression at single degree prevision (Fig. S8). Results (Fig. 3d) shows that the total aerosol water decreases by 12.7% when nitrates are reduced to the observational level, indicating that the hygroscopic growth is too large due to the overestimated nitrate. We add comment at Line 446: "The total aerosol water concentration decreases by 12.7% in GC_2m_PBLH_NIT from GC_2m_PBLH as nitrate is reduced."

Line 261 – Please provide references for this statement. "Vertical concentration gradients within 120 m of the surface have been widely observed for aerosol species in previous field campaigns."

Thanks. Provided in the revised manuscript at Line 327: "Vertical concentration gradients within 120 m of the surface have been widely observed for aerosol species in previous field campaigns (Sievering et al., 1994; Prabhakar et al., 2017; Franchin et al., 2018)."

Line 265 – How different is the temperature and RH between 2m and the lowest model level, and how would that impact thermodynamic partitioning of sulfate-nitrate-ammonium aerosol?

Based on the GEOS-FP dataset, the averaged RH over the US at the lowest model level is 0.66, which is 8.7% lower than that at 2m. The averaged T over the US at the lowest model level is 287.7 K, which is 0.02% higher than that at 2m. To investigate the impacts on thermodynamic partitioning of sulfate-nitrate-ammonium (SNA) aerosol, we conducted E-AIM (Extended AIM Aerosol Thermodynamics Model) simulations (Wexler and Clegg, 2002) at 2m and the lowest model level with RH&T inputs from GEOS-FP and SNA composition inputs from the GC_2m (Table 1). The RH, T and SNA composition inputs were all averages over the continental US. Results of the E-AIM simulations showed that the impacts of RH&T differences between the lowest model level and 2m on thermodynamic partitioning of SNA aerosol is overall insignificant. Higher RH at 2m can lead to SNA aerosol transit from solid form to aqueous form and can slightly increase the ratio (<5%) of the partitioned aerosol phase in the SNA system, which usually occurs overnight.

At Line 350 in the revised manuscript, we stated that "The correction method described by Eq. 6 and Eq. 7 does not account for the impacts of relative humidity (RH) and temperature (T) differences between the lowest model level and 2m on thermodynamic partitioning of sulfate-nitrate-ammonium (SNA) aerosol. Nevertheless, by conducting simulations of the Extended AIM Aerosol Thermodynamics Model (Wexler and Clegg, 2002) using GEOS-FP relative humidity (RH), Temperature (T) and GC_2m SNA composition at 2m and the lowest model level, we found the impacts are overall insignificant. Higher RH at 2m leads to SNA aerosol transition from solid to aqueous form and only slightly increases the ratio (<5%) of the partitioned aerosol phase in the SNA system, which usually occurs overnight."

Line 306 – Please briefly describe how the AMDAR PBLH is generated.

Thanks for the suggestion. A brief description of the estimation method of AMDAR PBLH is added in the revised manuscript at Line 378: "The AMDAR PBLH is defined as the lowest level at which the bulk Richardson number exceeds a critical value of 0.5 (Zhang et al., 2020). The vertically resolved bulk Richardson number is calculated from vertical profiles of temperature, humidity and wind speed in the AMDAR dataset."

Line 313 – Please comment more on why the PBLH in GEOS-FP might be too high and what that could mean for other parts of the simulation (e.g., temperature).

Thanks for pointing this out and suggesting thinking more about temperature. Comparisons of GEOS-FP and ceilometer observations in Millet et al., (2015) showed that GEOS-FP significantly overestimated the temperature at 2 meters during daytime. The diel patterns in the overestimation of

PBLH and temperature at 2 meters well matched to each other in GEOS-FP. This indicates that surface heating in GEOS-FP might be too strong leading to the excessive daytime PBLH.

In the revised manuscript, we added comments at Line 386: "The daytime overestimation in GEOS-FP PBLH is most likely due to excessive surface heating in the dataset. As reported in Millet et al., (2015), the daytime temperature at 2 meters in GEOS-FP was notably higher than that observed by ceilometer and the diel pattern of the bias in GEOS-FP temperature at 2 meters well matched that in PBLH."

Line 315 – Could you explain why it would be reasonable to have a PBLH maximum in spring?

Thanks for the suggestion. As stated at line 384 in the manuscript, PBLH is positively related to not only thermal radiation received by Earth's surface but also to surface wind speed (Guo et al., 2016). According to the AMDAR dataset (Zhang et al., 2020), the averaged surface wind speed reached 3.0 m/s over the continental US in Spring 2016, which was 13.2% higher than that in summer 2016. With stronger wind at the surface in spring, turbulence and vertical mixing became more developed, leading to higher PBLH. Besides the AMDAR dataset, daytime boundary layer heights in US (Seidel et al., 2010) and Chinese cities (Guo et al., 2016) estimated from radiosonde observations also showed maxima in Spring.

To make these clearer, we explain these at Line 389 in the revised manuscript: "The average daytime AMDAR PBLH reaches a maximum in spring, slightly higher than that in summer, likely reflecting greater surface wind speed in spring than in summer according to the AMDAR observations, leading to greater turbulence and vertical mixing, and higher PBLH. GEOS-FP PBLH exhibits much higher values in summer than in spring. This inconsistency might be caused by stronger overestimation of GEOS-FP PBLH in summer introduced by excessive surface heating in the GEOS-FP dataset (Millet et al., 2015)."

Line 384 – See the additional two papers on nitrate overestimates (above Line 142). It would be worth commenting on the reasons described in those papers. The authors should comment that a large bias remains in their modeled nitrate. For example, there seems to be general consensus that $NO_2$ hydrolysis shouldn't be very important in the model and that $N_2O_5$ hydrolysis should be the main pathway (one example https://agupubs.onlinelibrary.wiley.com/doi/full/10.1029/2019GL085498). Which pathways dominate the authors' results? The authors could consider turning $NO_2$ hydrolysis off if it seems unreasonably large.

Thanks for the suggestion. In the revised manuscript, we state that large biases remain in our simulations and comment on the reasons described in the recommended papers (Miao et al., 2020; Travis et al., 2022). Specifically, the reasons mentioned in these papers are uncertainties in the $N_2O_5$ and $NO_2$ aerosol uptake coefficients (Miao et al., 2020), underestimated dry deposition of $HNO_3$ and overestimated nitrate production from $NO_2$ hydrolysis introduced by an overly shallow nighttime mixing layer (Travis et al., 2022). As shown in Travis et al., (2022), implementing 5 times greater $HNO_3$ dry deposition and revised uptake coefficients for $NO_2$ and $N_2O_5$ helped to reduce nitrate positive biases, but the biases from midnight to early morning remains.

In terms of chemical pathways, we followed Travis et al., (2022) and added extra sensitivity simulations (Sect. S3) in the revised manuscript. We found the dominant pathway of nighttime nitrate is $N_2O_5$ hydrolysis in our simulations. By turning off $N_2O_5$ hydrolysis on aerosols (GC_Chem_S2 in Table S3) we found the simulated nighttime nitrate was significantly reduced and the resultant diel $PM_{2.5}$ matched the FEM observations with high consistency (Fig. S9). While turning off $NO_2$ hydrolysis had much

smaller impacts on modeled nitrate (Fig. S10) and did not help to improve the resultant diel $PM_{2.5}$ variation (Fig. S9). Notably, according to Fig. 6, GEOS-FP PBLH agrees well with the observation-based AMDAR PBLH during the night, without notable underestimation as identified in Travis et al., (2022) over Seoul, Korea.

We added the above discussion in the revised manuscript as below.

Sect. S3 added to the supplement: "Following Travis et al., (2022), four additional sensitivity simulations (Table S3) were designed based on GC_2m (Table 1) for identifying primary chemical pathways of nighttime $PM_{2.5}$ nitrate in GEOS-Chem. As shown in Table S3, GC_Chem_S1 turns off $NO_2$ hydrolysis (reaction R1), GC_Chem_S2 turns off $N_2O_5$ hydrolysis (reaction R2&R3), GC_Chem_S3 turns off $NO_3$ hydrolysis (reaction R4), and GC_Chem_S4 turns off all nighttime nitrate chemistry in GEOS-Chem (reaction R1-R4) ... ..."

Sect. 5.6 "Impacts from nitrate aerosols" is added at Line 425: "In Fig. 3b, hourly emissions reduce nighttime concentrations of nitrate and organics, primarily reflecting diminished nighttime emissions of $NH_3$, $NO_x$, and organic carbon. Accounting for vertical representativeness further reduces nighttime concentrations of nitrate (Fig. 3c), leading to reduced positive biases of 24h-averaged nitrate mass against in situ observations (Fig. S8). Nevertheless, positive nitrate biases remain in the GC_2m_PBLH simulation (Fig. S8), which has been a long-standing issue in GEOS-Chem (Heald et al., 2012; Zhu et al., 2013). According to recent works (Miao et al., 2020; Zhai et al., 2021; Travis et al., 2022), uncertainties in aerosol uptake coefficient for $N_2O_5$ and $NO_2$, underestimated dry deposition of $HNO_3$ and overly shallow nighttime mixing layer are possible contributors. But none of these fully resolve the diel biases of nitrates in GEOS-Chem, indicating the biases are likely caused by misrepresentation in both chemistry and meteorology in the model. Following analyses by Travis et al. (2022) over Seoul, Korea, we conducted sensitivity simulations (Sect. S3) and found $N_2O_5$ hydrolysis dominates the nighttime nitrate production (Fig. S9 and Fig. S10) in our simulations over the US, which is consistent with a previous work (Alexander et al., 2020). As shown in Fig. S9, turning off $N_2O_5$ hydrolysis largely reduces the $PM_{2.5}$ biases from midnight to early morning and yields diel $PM_{2.5}$ variation highly consistent with observations. The bias in simulated nitrate mass concentrations is also reduced by turning off $N_2O_5$ hydrolysis (Fig S8). The results indicate the $N_2O_5$ hydrolysis overnight might be excessive in the model. It is also possible that the performance of the simulation without $N_2O_5$ hydrolysis on aerosols is an indicator of multiple chemical and physical processes affecting nitrate as explored by Miao et al. (2020), Zhai et al. (2021) and Travis et al. (2022). While the full origins of the GEOS-Chem nitrate bias remain unknown, we examine the effects on $PM_{2.5}$ of constraining nitrate concentrations by developing the GC_2m_PBLH_NIT simulation, in which the modeled nitrate concentrations are halved from GC_2m_PBLH to better represent the US average of in situ observations (Fig. S8). The bias of the diel amplitude of $PM_{2.5}$ in GC_2m_PBLH_NIT against FEM observations is reduced to -12% (Fig. 2). The total aerosol water concentration decreases by 12.7% in GC_2m_PBLH_NIT from GC_2m_PBLH as nitrate is reduced. These results motivate further investigation of the nitrate bias in GEOS-Chem."

At Line 504: "Nitrate appears to be an important contributor to the bias, which is not fully understood as discussed in Sect. 5.6. Insufficient vertical and horizontal resolution in our simulations to fully resolve nocturnal stratification and horizontal source separation (Zakoura and Pandis, 2018; Boys 2022) are possible contributors."

Line 425 – Could you be quantitative about the impact of resolution?

Thanks for the suggestion. At finer resolution, the excessive $PM_{2.5}$ accumulation at night in GEOS-Chem can be reduced. To be quantitative, we re-expressed this improvement by providing percentage $PM_{2.5}$ increases from 18:00 LT to 6:00 LT in both observation and simulations at Line 519: "Comparison of the GEOS-Chem simulations at 0.25° × 0.3125° and 2° × 2.5° against the FEM observations (Fig. S12) reveals that higher spatial resolution better enables the model to reproduce the observed diel $PM_{2.5}$ variation through reducing the excessive $PM_{2.5}$ accumulation during nighttime (18:00 LT- 6:00 LT). At the coarse spatial resolution, the simulated $PM_{2.5}$ mass concentrations increase by 36.3% from 18:00 LT to 6:00 LT, greater than the observed 5.8% increase. At the finer spatial resolution, that nighttime increase of $PM_{2.5}$ mass concentrations reduces to 20.3%."

Line 456 – What is this product and what will the increased resolution be in the PBL?

Thanks for pointing this out. It's GEOS-6, which was planned to have a doubled number of vertical levels within PBL (NASA, 2012) compared to GEOS-FP. To make this clearer, we revised the text at Line 562 as "The importance of vertical resolution in representing $PM_{2.5}$ diel variation identifies an advantage to be offered by a forthcoming GEOS-6 dataset with a planned doubled number of vertical levels in the PBL compared to GEOS-FP (NASA, 2012)."

References:

Emerson, E. W., Hodshire, A. L., DeBolt, H. M., Bilsback, K. R., Pierce, J. R., McMeeking, G. R., & Farmer, D. K. (2020). Revisiting particle dry deposition and its role in radiative effect estimates. Proceedings of the National Academy of Sciences, 117(42), 26076-26082.

Environmental Protection Agency. (2007, April). Guidance on the Use of Models and Other Analyses for Demonstrating Attainment of Air Quality Goals for Ozone, $PM_{2.5}$, and Regional Haze. Retrieved from https://www.epa.gov/sites/default/files/2020-10/documents/final-03-pm-rh-guidance.pdf

Environmental Protection Agency. (2021, March). Standard Operating Procedure for the Continuous Measurement of Particulate Matter for Thermo Scientific TEOM 1405-DF Instrument [PDF file]. Retrieved from https://www.epa.gov/sites/default/files/2021-03/documents/905505_teom_sop_draft_final_sept09.pdf

Environmental Protection Agency. (2023, June). List of Federal Reference Method (FRM) and Federal Equivalent Method (FEM) Designations (June 2023) [PDF file]. Retrieved from https://www.epa.gov/system/files/documents/2023-06/List_of_FRM_FEM_%20June%202023_Final.pdf

GEOS-Chem Aerosols Working Group. (2021, November). Definitions of $PM_{2.5}$ and $PM_{10}$ for GEOS-Chem. GEOS-Chem Wiki. Retrieved from http://wiki.seas.harvard.edu/geos-chem/index.php/Particulate_matter_in_GEOS-Chem

Guo, J., Miao, Y., Zhang, Y., Liu, H., Li, Z., Zhang, W., ... & Zhai, P. (2016). The climatology of planetary boundary layer height in China derived from radiosonde and reanalysis data. Atmospheric Chemistry and Physics, 16(20), 13309-13319.

Miao, R., Chen, Q., Zheng, Y., Cheng, X., Sun, Y., Palmer, P. I., ... & Zhang, Y. (2020). Model bias in simulating major chemical components of $PM_{2.5}$ in China. Atmospheric Chemistry and Physics, 20(20), 12265-12284.

Millet, D. B., Baasandorj, M., Farmer, D. K., Thornton, J. A., Baumann, K., Brophy, P., ... & Xu, J. (2015). A large and ubiquitous source of atmospheric formic acid. Atmospheric Chemistry and Physics, 15(11), 6283-6304.

National Aeronautics and Space Administration (NASA). (2012). A Brief Summary of Plans for the GMAO Core Priorities and Initiatives for the Next 5 years [PDF file]. Retrieved from https://gmao.gsfc.nasa.gov/docs/GMAO_Summary.pdf

Seidel, D. J., Ao, C. O., & Li, K. (2010). Estimating climatological planetary boundary layer heights from radiosonde observations: Comparison of methods and uncertainty analysis. Journal of Geophysical Research: Atmospheres, 115(D16).

Thermo Fisher Scientific. (2013, Jan). EPM Manual Model 5030 Sharp [PDF file]. Retrieved from https://tools.thermofisher.com/content/sfs/manuals/EPM-manual-Model%205030%20SHARP.pdf

Travis, K. R., Crawford, J. H., Chen, G., Jordan, C. E., Nault, B. A., Kim, H., ... & Kim, M. J. (2022). Limitations in representation of physical processes prevent successful simulation of $PM_{2.5}$ during KORUS-AQ. Atmospheric Chemistry and Physics, 22(12), 7933-7958.

Wexler, A. S., & Clegg, S. L. (2002). Atmospheric aerosol models for systems including the ions $H^+$, $NH_4^+$, $Na^+$, $SO_4^{2-}$, $NO_3^-$, $Cl^-$, $Br^-$, and $H_2O$. *Journal of Geophysical Research: Atmospheres*, *107*(D14), ACH-14.

Zhang, Y., Sun, K., Gao, Z., Pan, Z., Shook, M. A., & Li, D. (2020). Diurnal climatology of planetary boundary layer height over the contiguous United States derived from AMDAR and reanalysis data. Journal of Geophysical Research: Atmospheres, 125(20), e2020JD032803.